# Phage-microbe dynamics after sterile faecal filtrate transplantation in individuals with metabolic syndrome: a double-blind, randomised, placebo-controlled clinical trial assessing efficacy and safety

Koen Wortelboer [1,2,3,7], Patrick A. de Jonge[1,2,3,7], Torsten P. M. Scheithauer[1,2,3], Ilias Attaye[2,3,4], E. Marleen Kemper [1,5], Max Nieuwdorp [1,2,3,4,6] & Hilde Herrema [1,2,3] ✉

Bacteriophages (phages) are bacterial viruses that have been shown to shape microbial communities. Previous studies have shown that faecal virome transplantation can decrease weight gain and normalize blood glucose tolerance in diet-induced obese mice. Therefore, we performed a double-blind, randomised, placebo-controlled pilot study in which 24 individuals with metabolic syndrome were randomised to a faecal filtrate transplantation (FFT) from a lean healthy donor ($n = 12$) or placebo ($n = 12$). The primary outcome, change in glucose metabolism, and secondary outcomes, safety and longitudinal changes within the intestinal bacteriome and phageome, were assessed from baseline up to 28 days. All 24 included subjects completed the study and are included in the analyses. While the overall changes in glucose metabolism are not significantly different between both groups, the FFT is well-tolerated and without any serious adverse events. The phage virion composition is significantly altered two days after FFT as compared to placebo, which coincides with more virulent phage-microbe interactions. In conclusion, we provide evidence that gut phages can be safely administered to transiently alter the gut microbiota of recipients.

The metabolic syndrome (MetSyn) constitutes a major global health concern[1]. This combination of clinical manifestations that are associated with insulin resistance affects nearly a quarter of the world population and increases the risk for cardiometabolic disease, such as type 2 diabetes (T2D) and cardiovascular disease[2,3]. The intestinal microbiota are increasingly seen as contributors to these diseases, e.g., through production of certain microbial metabolites and induction of low-grade inflammation[4,5].

[1]Amsterdam UMC location University of Amsterdam, Experimental Vascular Medicine, Amsterdam, The Netherlands. [2]Amsterdam Cardiovascular Sciences, Diabetes & Metabolism, Amsterdam, The Netherlands. [3]Amsterdam Gastroenterology Endocrinology Metabolism, Endocrinology, metabolism and nutrition, Amsterdam, The Netherlands. [4]Amsterdam UMC location University of Amsterdam, Vascular Medicine, Amsterdam, The Netherlands. [5]Amsterdam UMC location University of Amsterdam, Department of Pharmacy and Clinical Pharmacology, Amsterdam, The Netherlands. [6]Amsterdam UMC location Vrije University Medical Center, Department of Internal Medicine, Diabetes Center, Amsterdam, The Netherlands. [7]These authors contributed equally: Koen Wortelboer, Patrick A. de Jonge. ✉e-mail: h.j.herrema@amsterdamumc.nl

Previously reported microbial effects on human health have been mainly attributed to the bacterial component of the microbiota[6]. However, the gut microbiome is an ecosystem, which, in addition to bacteria, contains viruses, archaea, fungi and protists[7]. The viral component predominantly comprises bacteriophages (98%)[8], which are present in similar numbers as bacteria in the gut[9]. Bacteriophages (phages from hereon) are bacterial viruses that exclusively infect bacteria and, by doing so, often either kill bacteria (lysis) or incorporate themselves into the bacterial genome (lysogeny)[10]. Consequently, phages shape microbial communities in many ecosystems[11,12]. Moreover, phages have been implicated in human (gastrointestinal) disease[13–16], including diabetes[17,18]. We recently described decreased richness and diversity of the gut phageome in MetSyn, together with a larger inter-individual variation and altered composition[19].

Considering their ability to modulate gut bacteria and their function[20], phages are of special interest in ongoing endeavours to alter the human gut microbiome to benefit human health. Furthermore, the emergence of multidrug-resistant bacteria has led to an increasing interest in phage therapy, in which host-specific phages target specific pathogenic bacteria without affecting the commensal microbiota[21,22]. Such phage cocktails can be very effective in treating monoclonal bacterial infections, but are in general not sufficient to (beneficially) alter a complete microbiome[23,24]. Therefore, there is growing interest in the transfer of virus-like particles (VLP) isolated from the faecal microbiota, generally called a faecal virome transplantation (FVT). In mice, it has been shown that FVT induced a comparable effect as a faecal microbiota transplantation (FMT), in which the complete faecal microbiota of a healthy donor is transferred[25,26]. Moreover, in a small human pilot study, an FMT depleted of bacteria, also known as a sterile faecal filtrate transplantation (FFT), was successful in curing five individuals from a recurrent *Clostridioides difficile* infection[27]. Compared to FMTs, an FFT or FVT depleted of living microorganisms has a lower risk of transferring unknown pathogenic bacteria, which might improve safety.

Modulation of gut microbiota composition through FMT has been shown to improve peripheral insulin sensitivity in individuals with MetSyn[28,29]. Moreover, an FVT from lean donor mice was able to decrease weight gain and normalise blood glucose tolerance in diet-induced obese mice[30]. This effect was likely mediated through alterations in the gut microbiota induced by phages, as prior treatment with antibiotics disrupted the bacterial hosts and thereby counteracted the effect of the FVT. This raised the question whether the transfer of faecal phages could induce a similar effect as FMT in human individuals with MetSyn.

To study the effect of faecal phages on glucose metabolism, comparing a clean and concentrated faecal virome transplant with a phage-inactivated transplant would be most desirable. Unfortunately, the IRB only allowed us to minimally process the faecal suspension that is usually used for FMT, so we chose an FFT approach. We were hence not able to remove components other than bacteria from the filtrate. However, since phages are self-propagating entities with presumed longer effects on the microbial ecosystem than a single administration of metabolites, peptides or debris, we considered it justified to use the FFT to study phage-bacteria interactions and subsequent effects on glucose metabolism.

In this double-blind, randomised, placebo-controlled pilot study, we provide proof of concept that a faecal filtrate from lean healthy donors containing gut virions can be safely administered to MetSyn recipients. Moreover, gut phages have the potential to improve glycemic variability and alter phage-microbe dynamics. Although follow-up studies with cleaner, better defined and better-matched donor-recipient pairs are needed, this study provides a critical basis to do so and move the field forward.

## Results

### Inclusion of subjects and donors

To study whether an FFT could induce a similar effect on glucose metabolism as an FMT, we set up a prospective double-blind, randomised, placebo-controlled pilot study (Fig. 1a). Changes in glucose metabolism between day 0 and 28 were determined by the total area under the curve (AUC) for glucose excursion during an oral glucose tolerance test (OGTT), the primary outcome. Based on previous data from our group[28,29] and our hypothesis that a faecal phage transplant can be equally effective as an FMT[25–27,30], a sample size of 12 patients per group was deemed necessary.

Starting from October 2019, a total of 82 subjects signed the informed consent form and were screened, of whom 24 subjects were included and finished the study before December 2020 (Supplementary Fig. 1A). Most subjects were excluded because they did not have MetSyn according to the National Cholesterol Education Program (NCEP) criteria for the metabolic syndrome[31]. For the faeces donors, 24 subjects signed informed consent and were stepwise screened, resulting in 6 eligible donors (Supplementary Fig. 1B). Potential donors failed screening mainly due to carriage of parasites (11/24, 46%), followed by positive stool tests for pathogens (4/24, 17%) and exclusion based on questionnaire (3/24, 13%). Of these 6 eligible donors, only 3 (3/24, 13%) donated faeces for the production of a sterile faecal filtrate. Therefore, an additional 2 donors who were already actively donating for other FMT studies were included[32].

The 24 included MetSyn subjects were randomly assigned to receive an FFT (n = 12) or placebo (n = 12). As shown in Table 1, both groups were similar in baseline characteristics, such as age, sex, body mass index (BMI) and other MetSyn-associated parameters. Only the systolic blood pressure was significantly higher in the placebo group, although this difference disappeared at baseline and follow-up visits and was therefore probably a case of white coat hypertension during the screening. None of the individuals with MetSyn used concomitant medication and their diets were similar (Supplementary Table 1). Compared to the healthy donors, the MetSyn subjects differed, as expected, in almost every aspect of MetSyn-associated parameters (Table 1). All participant completed the follow-up visit at day 28.

### FFT is safe and well-tolerated

The FFT was well-tolerated by the participants and there were no serious adverse events. Compared to the placebo group, more subjects in the FFT group reported adverse events (AEs) that were likely or possibly related to the intervention (six vs two subjects, p = 0.19, Fisher's exact test), reporting in total more AEs (eight vs two AEs, p = 0.11, Fisher's exact test). The adverse events after FFT were in general mild gastrointestinal complaints, such as diarrhoea, constipation, bloating and nausea, which started in the days after the intervention (median 1 day, range 0–36) and disappeared after several days (median 3 days, range 0–27). Besides the transferred faecal phages, these adverse events could theoretically be induced through the transfer of eukaryotic or human viruses. However, as only 0.044 ± 0.3% (median: 0%) of reads mapped to such viruses, we could not ascertain whether these had an effect. To minimise negative effects from eukaryotic viruses, healthy stool donors were thoroughly screened for presence of known pathogenic viruses prior to donation.

Looking at the clinical safety parameters for liver and renal function, haematology and inflammation, we did not observe any differences between the FFT and placebo groups (Table 2). Interestingly, in both groups there was a significant increase in urea levels, which could be explained by the laxative that was used the evening prior to day 0, leading to less degradation of amino acids through the liver at baseline, and therefore less urea.

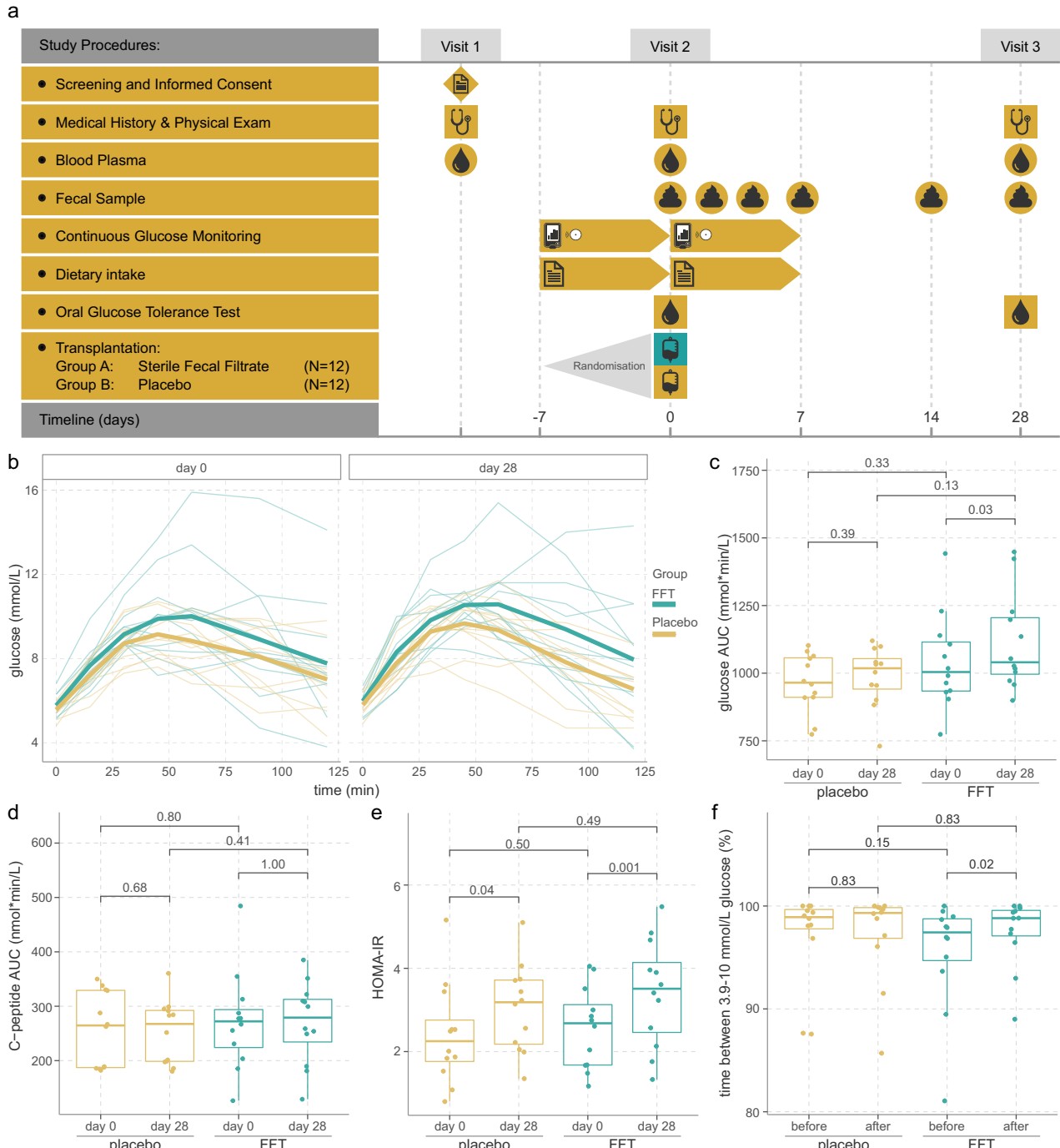

**Fig. 1 | Study overview and outcomes on glucose metabolism. a** Overview of the study. **b** Glucose excursions during the oral glucose tolerance test. One person who was randomised to the FFT group had progressed to type 2 diabetes, which was not apparent at the time of screening. **c** Total area under the curve (AUC) for glucose, and **d** for C-peptide did not significantly differ between the groups. Within both groups there was a small increase in glucose AUC between day 0 and 28, which was nominal significant within the FFT group, although this significance disappeared after correction for multiple testing. **e** Insulin resistance (HOMA-IR) measures did not significantly differ between the groups, but significantly increased from day 0 to day 28 in both groups. **f** Glucose variability, expressed as time between 3.9-10 mmol/L glucose, improved only within the FFT group between day 0 and day 28, which was nominal significant, but disappeared after correcting for multiple testing. Sample size for both FFT and placebo group is *n* = 12 subjects in all plots. Box plots show the median (middle line), 25th, and 75th percentile (box), with the 25th percentile minus and the 75th percentile plus 1.5 times the interquartile range (whiskers), and outliers (single points). P-values according to paired two-sided Wilcoxon signed-rank test. Source data are provided in the Source Data file.

## FFT improved glucose variability

Prior to the intervention and after 28 days at follow-up, subjects underwent an OGTT to assess their glucose metabolism (Fig. 1a), which was the primary outcome of the study. Glucose and C-peptide excursions during the OGTT at day 28 after the intervention were similar in the FFT and placebo group (Fig. 1b–d), as were within group alterations (day 0 vs day 28). In addition, we observed similar fasting glucose and insulin levels, insulin resistance (HOMA-IR) (Fig. 1e), and HbA1c values between the FFT and placebo group at day 28 (Table 3). Interestingly, we did observe a significant increase

**Table 1 | Baseline characteristics of MetSyn subjects and lean donors**

| | Placebo (n = 12) | Faecal filtrate (n = 12) | p-value | MetSyn combined (n = 24) | Donors (n = 5) | p-value |
|---|---|---|---|---|---|---|
| Age (years) | 49.3 (12.9) | 54.8 (8.9) | 0.20 | 52.0 (11.2) | 32.0 (7.81) | 0.003 |
| **Gender (n (%))** | | | 1.00 | | | 1.00 |
| Male | 7 (58.3%) | 7 (58.3%) | – | 14 (58.3%) | 3 (60.0%) | – |
| Female | 5 (41.7%) | 5 (41.7%) | – | 10 (41.7%) | 2 (40.0%) | – |
| **#MetSyn criteria (n (%))** | | | 0.38 | | | 0.000 |
| 1 | 0 | 0 | – | 0 | 3 (60.0%) | – |
| 2 | 0 | 0 | – | 0 | 2 (40.0%) | – |
| 3 | 4 (33.3%) | 7 (58.3%) | – | 11 (45.8%) | 0 | – |
| 4 | 5 (41.7%) | 4 (33.3%) | – | 9 (37.5%) | 0 | – |
| 5 | 3 (25.0%) | 1 (8.3%) | | 4 (16.7%) | 0 | – |
| BMI (kg/m²) | 36.1 (4.2) | 35.2 (5.7) | 0.76 | 35.7 (4.9) | 22.2 (1.9) | 0.000 |
| WHR | 0.98 (0.09) | 1.00 (0.06) | 0.49 | 0.99 (0.07) | 0.83 (0.07) | 0.001 |
| Systolic BP (mmHg) | 147 (17) | 131 (17) | 0.047 | 139 (18) | 124 (5) | 0.039 |
| Diastolic BP (mmHg) | 93 (10) | 90 (11) | 0.74 | 91 (11) | 76 (5) | 0.004 |
| Pulse (beats/min) | 66 (12) | 69 (11) | 0.77 | 68 (11) | 60 (11) | 0.17 |
| Glucose (mmol/L) | 5.6 (0.5) | 5.7 (0.5) | 0.99 | 5.7 (0.5) | 5.0 (0.4) | 0.007 |
| Insulin (nmol/L) | 85 (24) | 82 (28) | 0.83 | 84 (26) | 38 (12) | 0.001 |
| HOMA-IR | 2.91 (0.76) | 2.92 (1.02) | 1.00 | 2.91 (0.88) | 1.18 (0.37) | 0.000 |
| HbA1c (mmol/mol) | 36 (3.2) | 36 (4.3) | 0.75 | 36 (3.7) | 34 (2.1) | 0.19 |
| Cholesterol (mmol/L) | 5.30 (1.13) | 5.76 (1.53) | 0.63 | 5.53 (1.34) | 4.01 (0.45) | 0.015 |
| HDL (mmol/L) | 1.15 (0.22) | 1.22 (0.23) | 0.52 | 1.19 (0.22) | 1.52 (0.26) | 0.010 |
| LDL (mmol/L) | 3.43 (0.98) | 3.71 (1.25) | 0.89 | 3.57 (1.11) | 2.09 (0.58) | 0.003 |
| Triglycerides (mmol/L) | 1.60 (0.58) | 1.84 (0.75) | 0.59 | 1.72 (0.67) | 0.89 (0.29) | 0.007 |
| CRP (mg/L) | 4.0 (5.1) | 6.2 (7.3) | 0.25 | 5.1 (6.2) | 1.4 (1.0) | 0.07 |

Unless otherwise specified, data are reported as mean (SD). Statistical testing between the placebo and faecal filtrate groups and metabolic syndrome subjects and donors is performed by independent Mann–Whitney *U* test for continuous variables and by Chi-square test for categorical and binary variables. All tests were two-sided. Source data are provided in the Source Data file.
*MetSyn* metabolic syndrome, *BMI* Body Mass Index, *WHR* waist-hip ratio, *BP* blood pressure, *HOMA-IR* Homeostatic Model Assessment for Insulin Resistance, *HDL* high-density lipoprotein, *LDL* low-density lipoprotein, *CRP* C-reactive protein.

in fasted insulin levels and associated HOMA-IR values between day 0 and 28 within both the FFT and placebo group. However, when comparing these two measures between the screening visit and day 28, they were similar. We can only speculate that this drop in insulin levels and associated HOMA-IR value at the baseline visit resulted from the laxative use the day prior to the intervention. Other baseline characteristics remained stable after intervention and were similar between the FFT and placebo group, such as BMI, blood pressure and cholesterol (Table 3).

In addition to the OGGT, subjects wore a continuous glucose monitoring (CGM) device (Freestyle Libre) from one week prior till one week after intervention. Looking at the results from the CGM measurements, the FFT and placebo group showed overall similar glucose levels and glucose variability markers after intervention (Supplementary Table 2). However, within the FFT group we identified a nominal significant improvement from 95.5% to 97.5% in the time between 3.9 and 10 mmol/L glucose after intervention ($p = 0.02$, Wilcoxon signed rank test, Fig. 1f). This indicated an improvement in glucose variability within the FFT group in the week after intervention.

**Bacterial and viral diversity remain stable after FFT**
To assess the effect of the FFT on the bacteriome and phageome, we collected multiple faecal samples from baseline up to day 28, and performed whole genome shotgun (WGS) sequencing (Fig. 2a)[33]. The phage populations derived from this WGS fraction mainly consist of integrated prophages. To study phage virions, VLPs were isolated from the same faecal samples, lysed and the purified DNA was shotgun sequenced as previously described[19]. After combining all viral

sequences from WGS and VLP fractions, we clustered them at 95% similarity into viral populations (VPs), a level comparable to species in bacteria[34].

Analysis of beta diversity showed that both the VLP and WGS phageomes were indistinguishable between donor and MetSyn participants at baseline (VLP: $p = 0.725$, PERMANOVA, Supplementary Fig. 2A; WGS: $p = 0.672$, PERMANOVA, Supplementary Fig. 2B). While this defies our earlier findings[19], this is likely due to the highly individual-specific viromes and the relatively small size of our study. In this study, we directly compared the VLP and WGS phageomes within the same patient longitudinally, whereas previous studies predominantly used different cohorts to compare the VLP and WGS phageomes[8]. Notably, the VLP phageome was radically different from the WGS phageome ($p = 0.001$, PERMANOVA, Supplementary Fig. 2C).

Next, we looked at the effect of FFT on the bacterial and viral richness (Fig. 2b) and alpha diversity in MetSyn subjects (Fig. 2c). These were comparable throughout the study between the FFT and placebo intervention. Interestingly, in both groups the bacterial richness and α-diversity reduced slightly the first days after the intervention, which was resolved by days 14–28, though these decreases were non-significant ($p > 0.05$, Wilcoxon signed-rank test). A similar non-significant trend was observed for the richness and diversity of the WGS phageome, which consists mainly of prophages that could have been depleted with their bacterial hosts. In contrast, the richness of the VLP phageome increased slightly by day 2 in both groups, while the α-diversity decreased only in the placebo group, albeit non-significant ($p > 0.05$, Wilcoxon signed-rank test).

**Table 2 | Differences in clinical safety markers after intervention**

| | | Placebo (n = 12) | Faecal filtrate (n = 12) | p-value |
|---|---|---|---|---|
| # AEs (n (%)) | Total | 13 (44.8%) | 16 (55.2%) | – |
| Relatedness of AEs (n (%)) | Likely | 0 (0%) | 2 (12.5%) | 0.21 |
| | Possibly | 2 (15.4%) | 6 (37.5%) | – |
| | Unlikely | 4 (30.8%) | 4 (25%) | – |
| | Not | 7 (53.9%) | 4 (25%) | – |
| # Subjects with AE (n (%)) possibly or likely related | ≥1 AE | 2 (16.7%) | 6 (50%) | 0.19 |
| | No AE | 10 (83.3%) | 6 (50%) | – |
| Bilirubin (µmol/L) | Day 0 | 12 (6) | 15 (9) | 0.39 |
| | Day 28 | 9 (5) | 12 (12) | – |
| p-value | – | 0.030 | 0.16 | – |
| AF (U/L) | Day 0 | 76 (18) | 69 (16) | 0.17 |
| | Day 28 | 79 (16) | 68 (16) | – |
| GGT (U/L) | Day 0 | 22 (10) | 26 (11) | 0.30 |
| | Day 28 | 22 (12) | 26 (9) | – |
| ASAT (U/L) | Day 0 | 28 (8) | 28 (7) | 0.70 |
| | Day 28 | 27 (8) | 25 (7) | – |
| ALAT (U/L) | Day 0 | 29 (11) | 29 (10) | 0.85 |
| | Day 28 | 28 (10) | 27 (9) | – |
| Creatinine (µmol/L) | Day 0 | 85 (18) | 76 (15) | 0.20 |
| | Day 28 | 82 (13) | 75 (15) | – |
| eGFR (ml/min/1.73 m$^2$) | Day 0 | 81 (12) | 86 (6) | 0.32 |
| | Day 28 | 83 (9) | 85 (7) | – |
| Urea (mmol/L) | Day 0 | 4.3 (0.9) | 4.4 (1.1) | 0.65 |
| | Day 28 | 4.8 (0.9) | 5.1 (1.3) | – |
| p-value | – | 0.047 | 0.012 | – |
| Haemoglobin (mmol/L) | Day 0 | 8.5 (1.0) | 8.8 (0.8) | 0.74 |
| | Day 28 | 8.6 (0.9) | 8.6 (0.6) | – |
| Haematocrit (L/L) | Day 0 | 0.41 (0.04) | 0.42 (0.03) | 0.84 |
| | Day 28 | 0.42 (0.04) | 0.41 (0.03) | – |
| MCV (fL) | Day 0 | 86.0 (4.7) | 88.0 (2.7) | 0.25 |
| | Day 28 | 86.5 (4.5) | 88.1 (3.0) | – |
| Thrombocytes (x10$^9$/L) | Day 0 | 265 (87) | 259 (45) | 0.85 |
| | Day 28 | 263 (73) | 259 (48) | – |
| Leucocytes (x10$^9$/L) | Day 0 | 6.2 (1.4) | 6.2 (1.2) | 0.56 |
| | Day 28 | 5.8 (1.2) | 6.3 (1.4) | – |
| Eosinophils (x10$^9$/L) | Day 0 | 0.15 (0.07) | 0.12 (0.06) | 0.61 |
| | Day 28 | 0.16 (0.08) | 0.16 (0.11) | – |
| Basophils (x10$^9$/L) | Day 0 | 0.04 (0.01) | 0.03 (0.02) | 0.27 |
| | Day 28 | 0.04 (0.02) | 0.04 (0.02) | – |
| Neutrophils (x10$^9$/L) | Day 0 | 3.64 (1.14) | 3.83 (0.97) | 0.36 |
| | Day 28 | 3.25 (0.99) | 3.82 (1.13) | – |
| Lymphocytes (x10$^9$/L) | Day 0 | 1.83 (0.40) | 1.70 (0.42) | 0.61 |
| | Day 28 | 1.81 (0.29) | 1.77 (0.48) | – |
| Monocytes (x10$^9$/L) | Day 0 | 0.48 (0.11) | 0.50 (0.10) | 0.60 |
| | Day 28 | 0.47 (0.14) | 0.50 (0.08) | – |
| Immunoglobulins (x10$^9$/L) | Day 0 | 0.02 (0.01) | 0.02 (0.01) | 0.31 |
| | Day 28 | 0.02 (0.01) | 0.02 (0.01) | – |

Unless otherwise specified data are reported as mean (SD). Statistical testing for categorical variables of AE relatedness was done by Chi-square test and for the number of subjects with an AE Fisher's exact test was used. Mixed model analyses were used to assess differences between groups and timepoints, whereafter post hoc analyses were performed with Bonferroni correction. All tests were two-sided. The p-values in the right column shows the overall effect of treatment on the variable and only when significant, the adjusted p-values from the post hoc tests are shown. The p-values underneath variables indicate statistically significant differences between days 0 and 28 within a treatment group. Source data are provided in the Source Data file.
AE adverse event, AF alkaline phosphatase, ALAT alanine aminotransferase, ASAT aspartate aminotransferase, eGFR estimated glomerular filtration rate, GGT gamma-glutamyl transferase, MCV mean corpuscular volume.

**Table 3 | Changes in physical and metabolic variables after intervention**

| | Day | Placebo (n = 12) | Faecal filtrate (n = 12) | p-value |
|---|---|---|---|---|
| BMI (kg/m$^2$) | 0 | 35.8 (4.0) | 35.3 (5.6) | 0.75 |
| | 28 | 36.1 (3.9) | 35.3 (5.8) | – |
| WHR | 0 | 0.97 (0.09) | 0.99 (0.07) | 0.58 |
| | 28 | 0.97 (0.09) | 0.99 (0.08) | – |
| Systolic BP (mmHg) | 0 | 134 (15) | 130 (17) | 0.39 |
| | 28 | 134 (16) | 126 (17) | – |
| Diastolic BP (mmHg) | 0 | 88 (11) | 83 (14) | 0.37 |
| | 28 | 86 (9) | 84 (14) | – |
| Pulse (beats/min) | 0 | 66 (9) | 70 (13) | 0.33 |
| | 28 | 65 (10) | 70 (11) | – |
| Glucose (mmol/L) | 0 | 5.5 (0.4) | 5.8 (0.5) | 0.19 |
| | 28 | 5.7 (0.5) | 5.9 (0.5) | – |
| Insulin (nmol/L) | 0 | 71 (34) | 72 (26) | 0.76 |
| | 28 | 87 (30) | 93 (34) | – |
| p-value | – | 0.036 | 0.007 | – |
| HOMA-IR | 0 | 2.41 (1.21) | 2.57 (0.97) | 0.55 |
| | 28 | 3.05 (1.06) | 3.41 (1.28) | – |
| p-value | – | 0.047 | 0.008 | – |
| HbA1c (mmol/mol) | 0 | 36.8 (2.6) | 35.4 (4.7) | 0.53 |
| | 28 | 35.5 (2.5) | 35.0 (4.5) | – |
| Cholesterol (mmol/L) | 0 | 4.87 (0.79) | 5.38 (1.32) | 0.28 |
| | 28 | 4.92 (0.78) | 5.33 (1.15) | – |
| HDL (mmol/L) | 0 | 1.05 (0.20) | 1.14 (0.18) | 0.41 |
| | 28 | 1.15 (0.30) | 1.21 (0.17) | – |
| p-value | – | 0.005 | 0.064 | – |
| LDL (mmol/L) | 0 | 3.06 (0.77) | 3.43 (1.08) | 0.32 |
| | 28 | 3.03 (0.88) | 3.42 (0.99) | – |
| Triglycerides (mmol/L) | 0 | 1.68 (0.61) | 1.80 (0.61) | 0.96 |
| | 28 | 1.66 (0.88) | 1.56 (0.64) | – |
| CRP (mg/L) | 0 | 3.0 (2.6) | 5.1 (4.6) | 0.17 |
| | 28 | 2.9 (2.9) | 4.6 (3.4) | – |

Unless otherwise specified, data are reported as mean (SD). Mixed model analyses were used to assess differences between groups and timepoints, whereafter post hoc analyses were performed with Bonferroni correction. All tests were two-sided. The p-values in the right column shows the overall effect of treatment on the variable and only when significant, the adjusted p-values from the post hoc tests are shown. The p-values underneath variables indicate statistically significant differences between days 0 and 28 within a treatment group. Source data are provided in the Source Data file.
BMI Body Mass Index, WHR waist-hip ratio, BP blood pressure, HOMA-IR Homeostatic Model Assessment for Insulin Resistance, HDL high-density lipoprotein, LDL low-density lipoprotein, CRP C-reactive protein.

### Increase in new phages independent of the intervention

Since we expected transfer of donor phages to the recipients, we looked at the abundance of phages shared between donor and recipient before and after the FFT. Although not significant, after FFT the VPs shared with the donor within the WGS phageome increased up to day 14 (p = 0.2, Wilcoxon signed-rank test, Fig. 3a). For the VLP phageome we found an opposite effect, where the VPs shared with the donor decreased non-significantly after the FFT (p = 0.3, Wilcoxon signed-rank test, Fig. 3b). The broader effect of the FFT on the phageomes was determined by analysing the abundance of new phages that appeared after the FFT within the WGS phageome (Fig. 3c) and within the VLP phageome (Fig. 3d). In both groups the abundance of new phages increased over time and although not significant, this increase was slightly higher in the FFT group, especially in the VLP phageome on day 2 (p = 0.2, Mann–Whitney U test). These results seem to indicate that the phageomes were perturbed in both the placebo and FFT groups. It further shows that donor-derived phages, especially the VLPs, were either mostly immediately removed from the gut or their engraftment was balanced with the removal of pre-existing VPs shared with the donors.

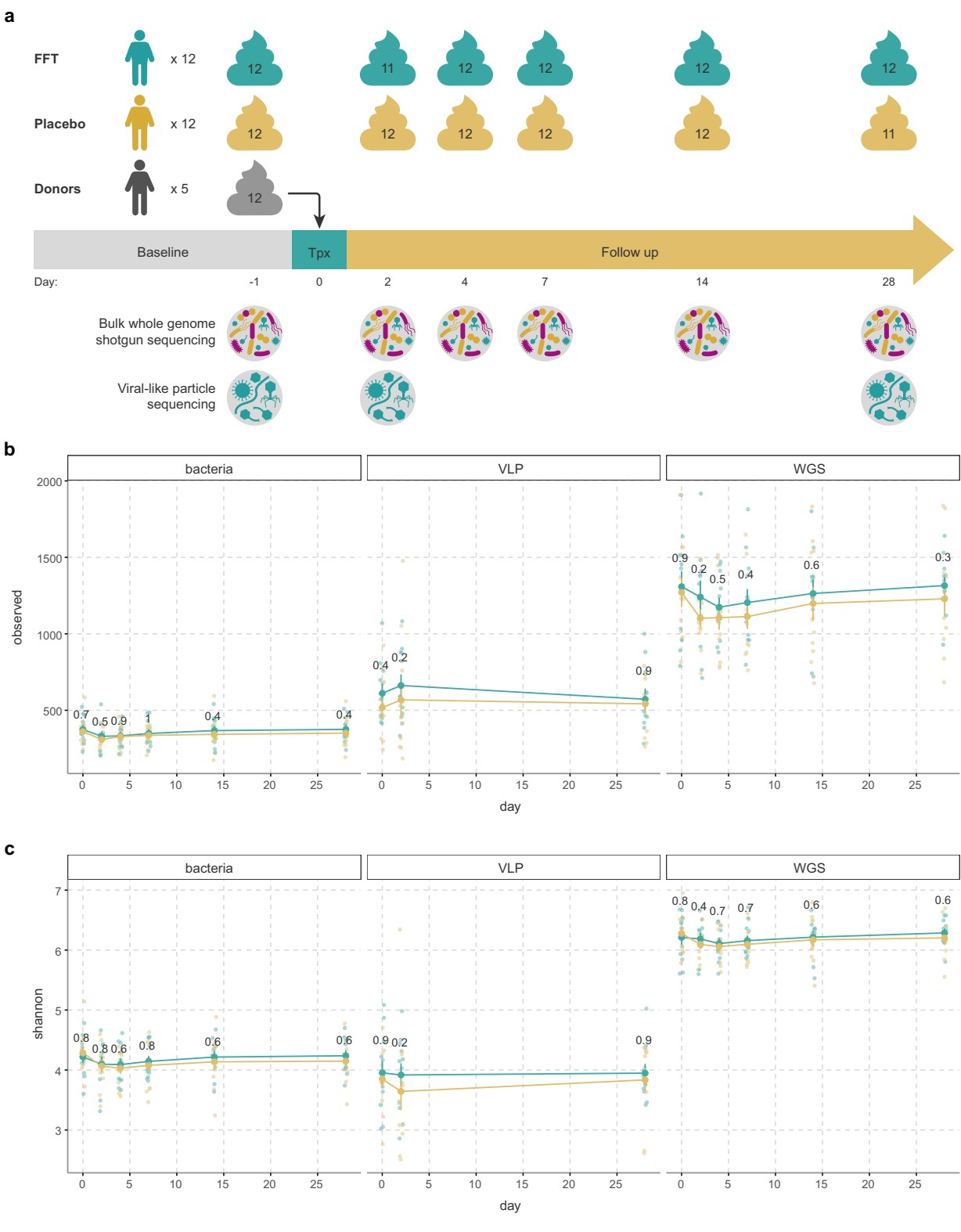

## FFT alters the phage composition of the VLP fraction

Subsequently, we looked at compositional changes within the bacteriome, WGS phageome and VLP phageome (Fig. 4a). Principal response curves showed no overall effect of the FFT on any of these communities compared to placebo, except for a significantly different composition of the VLP phageome on day 2 ($p = 0.02$, PERMANOVA). This difference in composition within the VLP phageome on day 2 was

also evident from a separate principal component analysis ($p = 0.028$, PERMANOVA, Fig. 4b). As this pointed toward a short-term effect of the FFT, we looked more specifically into VLP communities on day 2 and found 216 VPs that were differentially abundant between the FFT and placebo groups (Fig. 4c and Supplementary Data 1).

To get a better understanding of these phages, we looked at the bacterial host species that these differentially abundant VPs can infect.

**Fig. 2 | Overview of faecal samples sequenced, richness and diversity.**
**a** Overview of the faecal samples used for the bulk metagenomic sequencing (for bacteriome and phageome) and the metagenomic sequencing of the viral-like particles (VLP). **b** The richness (number of observed species) in the bacteriome, phage virions (VLP) and bulk-derived phageome (WGS) from baseline until follow-up at day 28. Though there were no significant differences between the placebo and faecal filtrate group, the richness in the bacteriome reduced slightly after both interventions. A similar trend was observed in the phageome (mostly prophages present in bacterial hosts), while the richness in the VLP fraction tended to increase

slightly at day 2 for both interventions. **c** The alpha-diversity (Shannon index) of the bacteriome, phage virions (VLP) and bulk-derived phageome (WGS) from baseline until follow-up at day 28. Again, no significant differences were found between the interventions. Similar to the richness, the diversity of the bacteriome and phageome slightly decreased directly after the interventions. For the free phages, the diversity decreased slightly in the placebo group, but not in the faecal filtrate group. Sample size for both FFT and placebo group is $n = 12$ subjects in all plots. Error bars represent the standard error of the mean. Source data are provided in the Source Data file.

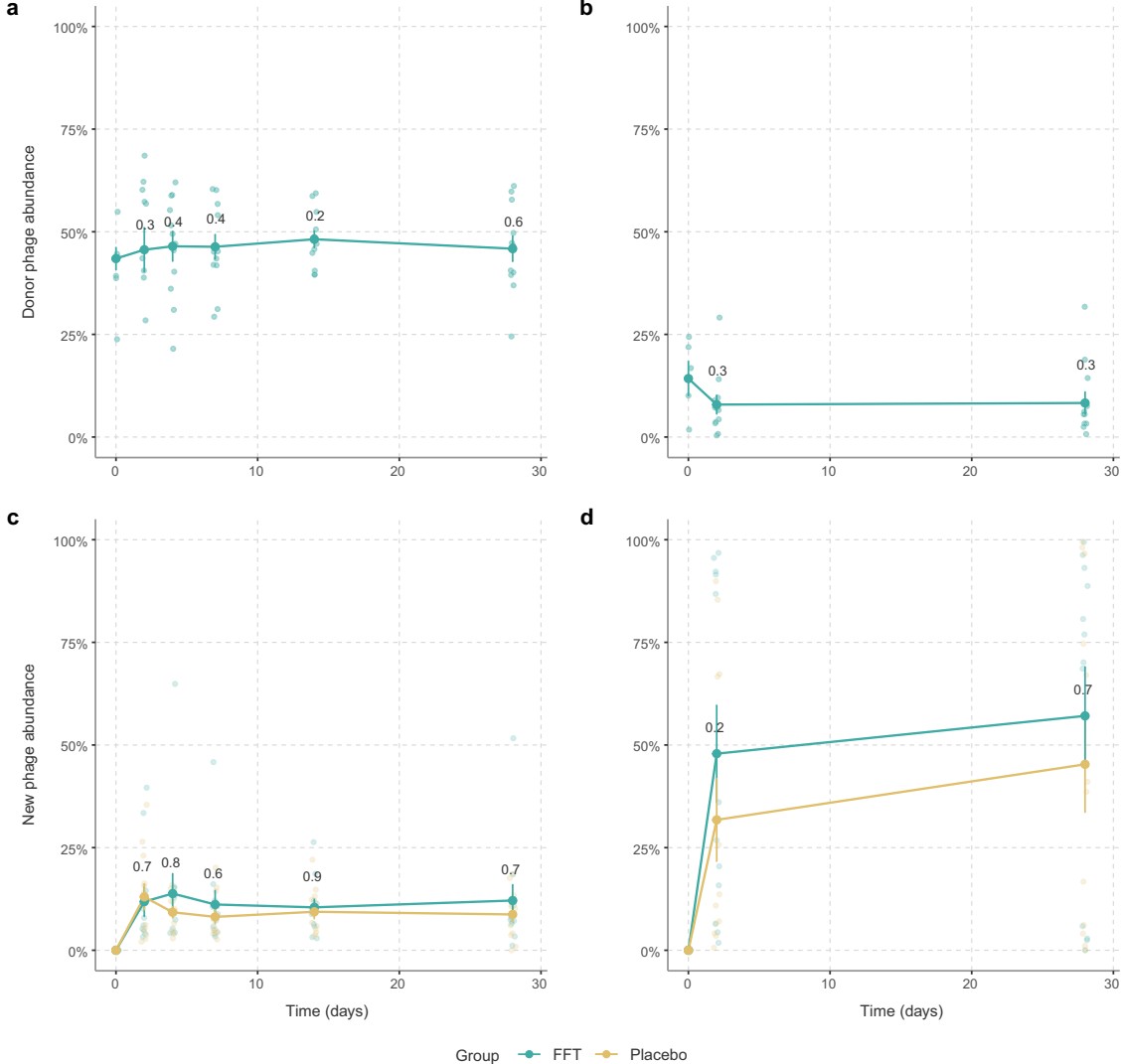

**Fig. 3 | Donor phage and new phage abundance in recipients. a** The percentage of phages that were shared between the donor and recipient within the phageome, and (**b**) within the phage virions after the faecal filtrate transplantation. There was a slight, non-significant increase in the relative abundance of (pro)phages shared with the donor after the intervention, while the relative abundance of phage virions that were shared with the donor slightly decreased. **c** The percentage of new

phages that were present after the intervention within the bulk-derived phageome, and (**d**) within the phage virions. In both, the relative abundance of new phages increased over time and although not significant, this increase was slightly higher in the FFT group. Sample size for both FFT and placebo group is $n = 12$ subjects in all plots. Error bars represent the standard error of the mean. Source data are provided in the Source Data file.

We observed 6 bacterial host species and 5 bacterial host genera of which the phages were significantly enriched among these VPs (Fig. 4d). The phages infecting some of these host bacteria, like *Roseburia intestinalis* and *Bacteroides* species, were differentially abundant in both FFT and placebo treatment groups. But others, like *Sutterella wadsworthensis* and *Scatocola faecigallinarum*, were notably exclusively differentially abundant in one of the two treatment groups. The only host species enriched among differentially abundant VPs and more prevalent in the placebo group was *S. wadsworthensis*, a

betaproteobacterium associated with gastrointestinal infections. Those more prevalent among the FFT group were taxonomically diverse, belonging to the *Bacteroidetes* (*Bacteroides spp.*), *Firmicutes* (*R. intestinalis*, *Faecousia* and *CAG-882*) and *Proteobacteria* (*S. faecigallinarum* and *CAG-267*).

## FFT induces an antagonistic phage-microbe interaction
Intrigued by the presence of differentially abundant VPs two days after FFT, we determined whether the dynamics between phages and their

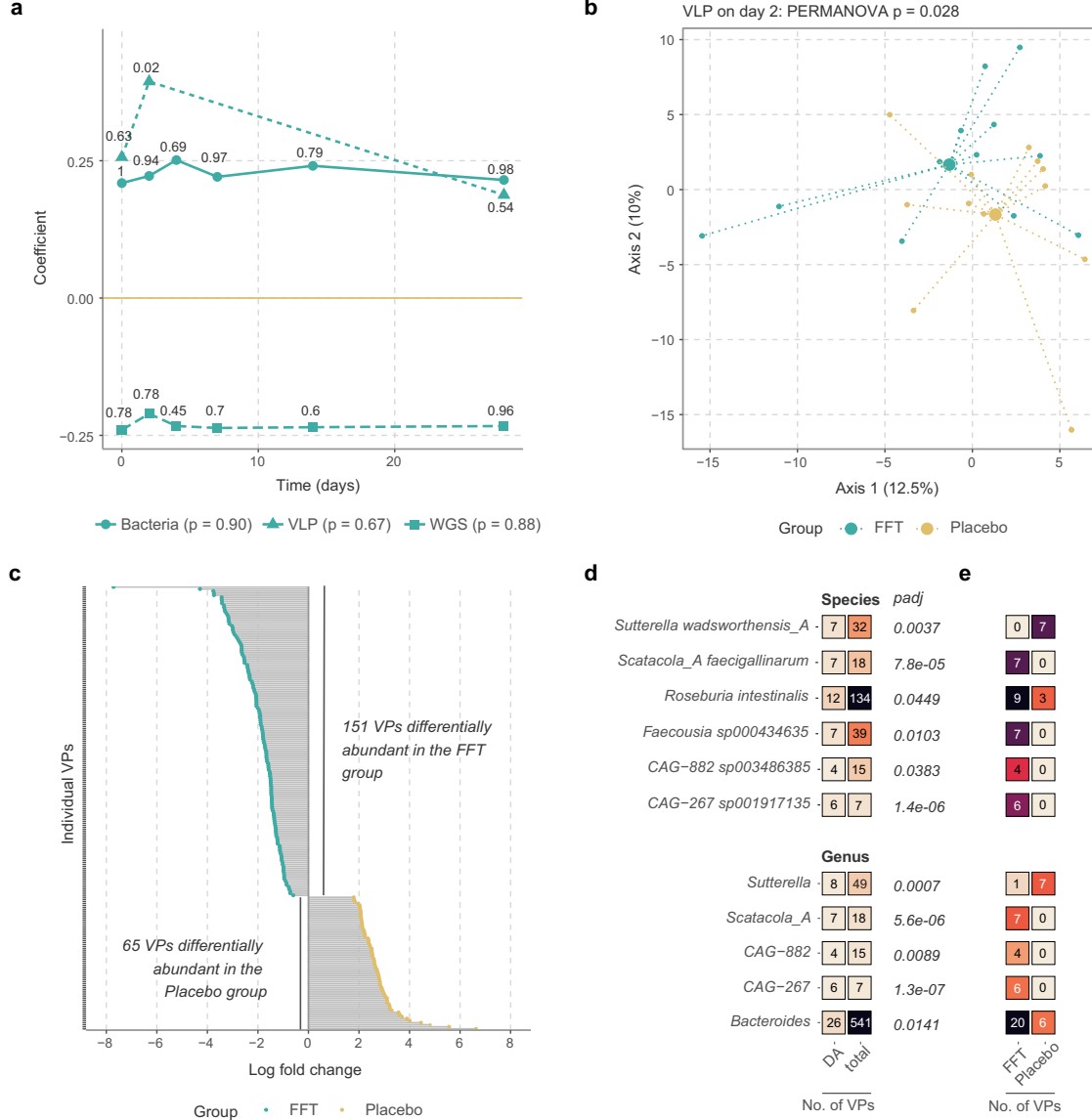

**Fig. 4 | Compositional differences between the FFT and placebo group.**
**a** Principal response curve showing how the FFT group (*n* = 12) differs from the placebo (*n* = 12, set to zero) in the bacteriome, bulk-derived phageome (WGS), and phage virion (VLP) composition. The coefficient is the canonical coefficient of treatment and significance in dispersion over time and at each separate time-point was calculated with permutation tests, corrected for multiple testing by the Benjamini-Hochberg method. **b** Principal component analysis of VLP composition after centred log-ratio transformation. Large points show the mean of each group (both *n* = 12). **c** Log fold change for all 216 viral populations (VP) indicated as

differentially abundant by ANCOM-BC analysis. For legibility, VP names are not shown, these can be found in Supplementary Data 1. **d** Bacterial host species of which the phages are enriched among differentially abundant VPs. The first column shows the number of differentially abundant VPs, the second the total number of VPs linked to a given host in the dataset, and padj shows the level of significance after testing for enrichment with a hypergeometric test, adjusted for multiple testing by the Benjamini-Hochberg method. **e** Splits up the data on the first column of D by participant group. Source data are provided in the Source Data file.

microbial hosts had changed. For this, we linked VPs to metagenome-assembled genomes (MAGs) from our WGS sequencing dataset and calculated the mean abundance change for all VP-MAG pairs belonging to a given species in a given sample. This showed opposing relationships between MAG and VP abundance in the first two days of the intervention (Fig. 5a): this was negatively correlated for the FFT group ($R = -0.13$, $p = 0.005$) and positively correlated in the placebo group ($R = 0.17$, $p = 0.000$). These results could indicate a difference in the ecological dynamics between the two sample groups, where the FFT group was dominated by lytic phage-bacterium interactions, while they were more likely to be lysogenic or chronic in the placebo group. These findings were unaltered when employing clr-transformed data. These effects intriguingly were less pronounced when comparing days

2 and 28 (FFT: $R = -0.043$, $p = 0.24$; placebo: $R = 0.12$, $p = 0.004$; Fig. 5b), and completely absent when comparing days 0 and 28 (Fig. 5c). Thus, the overall effect of the FTT on phage-host interactions seemed pronounced but short-lived.

## Discussion

In this randomised controlled clinical trial we administered a sterile faecal filtrate from healthy donors to human individuals with MetSyn. The FFT was well-tolerated and safe, with recipients experiencing solely mild gastrointestinal adverse effects. As the study group is small, larger studies with a longer follow-up are warranted to fully assess the safety profile of the FFT. However, compared to FMTs, an FFT depleted of living microorganisms has a lower risk of transferring unknown

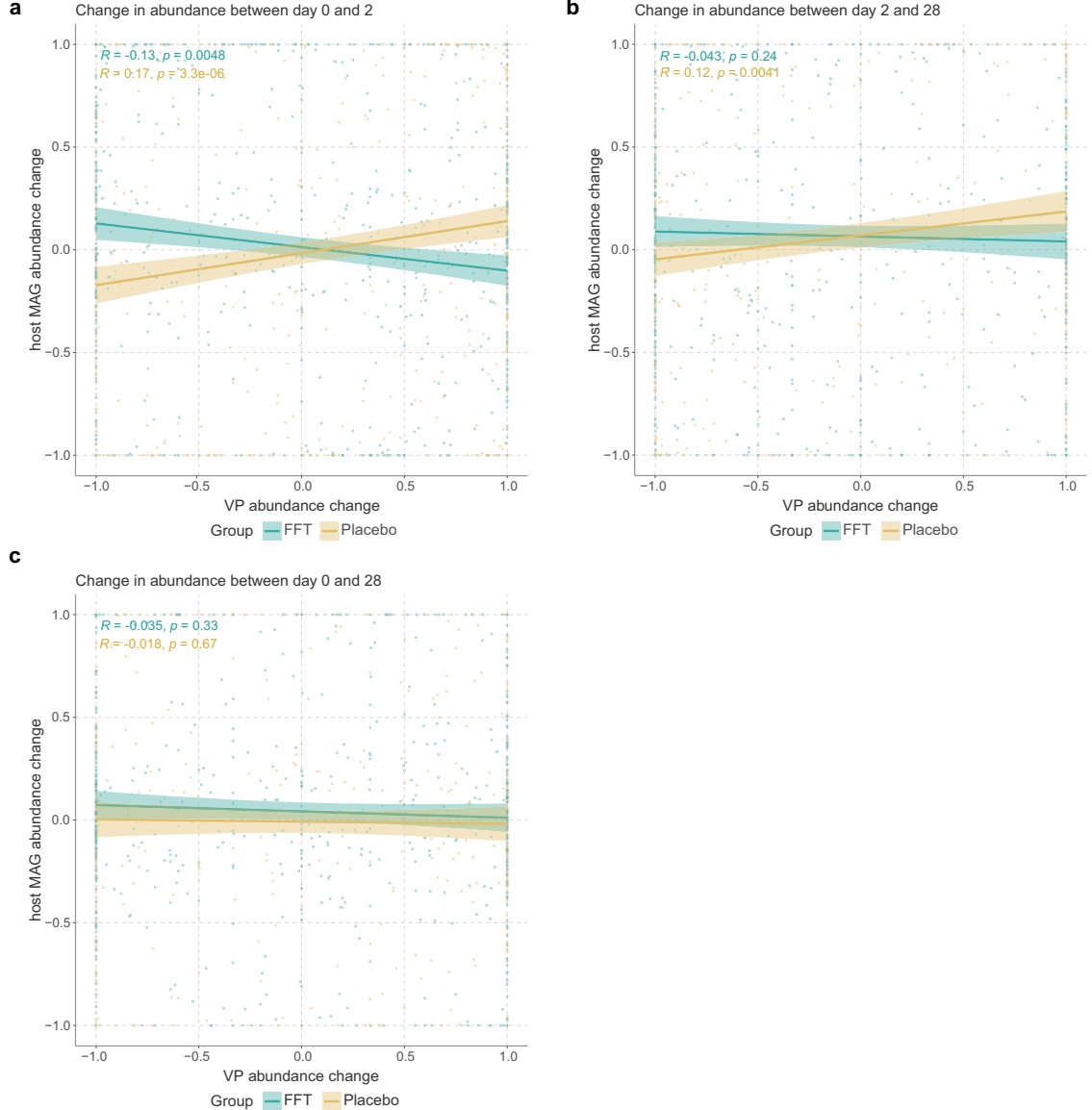

**Fig. 5 | Correlations between viral populations and their host bacteria.**
**a** Correlations plots showing the change in relative abundance between day 0 and day 2 of viral populations (VP) versus host bacteria. Each datapoint represents the interactions between the VPs and metagenome-assembled genomes (MAGs) of a particular species within a given sample. Linkages between VPs and MAGs were based on either CRISPR spacer hits or the presence of the VP as a prophage in the MAG. The Spearman correlation coefficient showed that phage-bacterium interactions in FFT samples tended toward antagonism, while those in placebo samples

were protagonistic. **b** Same as a, but showing the change in relative abundance between day 2 and day 28. The Spearman correlation coefficient showed that phage-bacterium interactions in placebo samples were protagonistic. **c** Same as a and b, but showing the change in relative abundance between day 0 and day 28. There was no overall correlation between changes in abundance of VPs and host MAGs. Shaded areas indicate 95% confidence interval. Significance according to two-sided Spearman's rank correlation coefficient. Source data are provided in the Source Data file.

pathogenic bacteria[27]. Since FMT has a good safety profile[35,36], this most likely holds true for FFTs as well. Compared to FMTs, it is relatively easier to further optimise and standardise FFTs, e.g., through lyophilization and encapsulation of faecal filtrate, as the viability of the many strict anaerobic bacteria does not have to be preserved. Such developments of FFT will ease the administration, reduce the invasiveness and provide an option for prolonged or maintenance therapy, even in a home-setting.

While we did find a slight improvement of the glucose variability in the FFT group, expressed as the time between 3.9 and 10 mmol/L glucose, the FFT and placebo groups showed similar glucose excursions during the OGGT performed at day 28. Previously, an FVT in diet-induced obese mice reduced weight gain and improved blood glucose tolerance[30]. However, FVTs differ slightly from FFTs, with phages being

more concentrated and washed to reduce bacterial debris, metabolites and antimicrobial peptides. Moreover, in this previous study, several donor phageomes were combined, resulting in a highly diverse phageome. In addition, compared to humans, microbiomes of mice are more similar due to the same housing and diet[37], thereby increasing the chance of highly specific bacteriophages encountering their host and, subsequently, modulating the microbiota. In humans, improvement of insulin sensitivity after lean healthy donor FMT in individuals with MetSyn has been reported[28,29]. These studies had a comparable study design as present study, with the major difference being the absence of the faecal bacteria in the intervention. Although this is not a direct comparison, we speculate that, in the case of MetSyn, the beneficial bacteria transplanted during an FMT significantly contribute to the improved glucose metabolism observed.

Nevertheless, the FFT was able to alter the phage virions or VLP phageome composition on day 2 compared to the placebo, showing 216 differentially abundant VPs. By day 28 this significant difference disappeared, which indicates the FFT effect was short-lived. Looking at the bacterial hosts of these differentially abundant phages, we found six host species that were significantly enriched, of which five were more prevalent in the FFT group. One of these bacterial hosts is the butyrate producer *Roseburia intestinalis,* which has been found to be depleted in MetSyn[38,39] and contributes to inflammatory signalling inhibition and intestinal barrier repair[40,41]. While the other bacterial species have not been directly linked to MetSyn previously, some of their relatives within the *Oscillospiraceae* (*Faecousia sp000434635*) and *Lachnospiraceae* (*CAG − 882 sp003486385*) have been implicated in obesity and MetSyn[42–45]. In line, the genus *Bacteroides* has been associated with obesity and MetSyn, both positively and negatively[46,47].

In addition, we speculate that the FFT induced virulent interactions between phages and their microbe hosts in the first two days after administration, while the phage-microbe interactions appeared more lysogenic/temperate in the placebo group. These virulent interactions in the FFT group could be the result of donor phages infecting and lysing the bacteria from the recipient. On the other hand, the introduction of novel donor phages could have induced the replication of existing prophages[48], thereby leading to more virulent interactions. As the number of previously unobserved VLP VPs increased on day 2, while donor-shared VLP VPs did not, we hypothesise the latter is more likely. It could be that some non-phage element of the FFT, such as fructose[49] or a phage-derived peptide[50], prompted integrated phages to excise from their bacterial hosts. Otherwise, it could also be that increased infection of bacteria by donor-derived phages caused lower bacterial abundance, resulting in higher phage lysis rates, in line with the piggyback the winner model of phage-host interactions[51]. Following this hypothesis further, growth of (some) bacterial species after the laxative treatment could have caused increased lysogeny among the phageomes in the placebo group.

Interestingly, changes in bacterial and viral diversity over time were similar between both groups. While we did observe a small, non-significant increase in the abundance of VPs shared with the donor in the WGS phageome, this abundance decreased non-significantly within the VLP phageome. This can in part be explained by the large increase in new phages within the VLP phageome (50-60%), which was bigger compared to the increase within the WGS phageome (-15%). This difference may have been caused by either the absence of low-abundance VPs in the WGS sequencing data due to their relatively smaller sizes, or a difference in community dynamics between VLP and WGS phageomes. The increase in new phages indicates that the phageome was perturbed, leading to an accelerated genomic recombination that stimulated phage evolution. However, since this happened in both groups, we hypothesise that this is, in part, an effect of the laxative pre-treatment. This laxative treatment could have removed pre-existing donor-shared VPs, and, by washing away part of the host bacteria, could have reduced the probability of donor phages infecting their host.

This study has several limitations. Although we did not find an overall effect on glucose metabolism, it is not possible to assess whether the FFT intervention was insufficient to alter the glucose metabolism or whether the effect is obfuscated by the small sample size and large heterogeneity within the MetSyn study population. Our sample-size for the current study was based on the assumption that the FFT would be as effective as an FMT in improving glucose metabolism[28,29], which is probably not the case. Unfortunately, based on current results where we observe a small non-significant increase in glucose AUC in both groups, it is not possible to repeat the power calculation. The increased glucose AUC could be seen as natural progression of MetSyn, but we speculate that this was caused by the laxative pre-treatment, which also reduced the fasting insulin levels

and associated HOMA-IR values at baseline. The laxative pre-treatment could also have reduced the FFT efficacy, by reducing the number of potential hosts for the transplanted phages. Therefore, for future studies with FFT, we would highly recommend to omit this step. In addition, pooling of donor phages and matching donors and recipients, thereby increasing the diversity and likelihood of a phage-host match, could further improve the efficacy of the FFT.

Due the ethical reasons, we had to keep the production of the bacteriophage transplant simple and straightforward, which is why we performed an FFT instead of an FVT in this human intervention study. Therefore, we cannot completely rule out any effect of other compounds present in the filtrate besides the phages, such as bacterial debris, metabolites and antimicrobial peptides. In line, we performed tangential flow filtration with sterile, single-use cassettes with a 0.2 μm membrane to reduce the potential risk of cross-contamination between donors. However, not all phages may pass through these pores and a pore size of 0.45 μm will result in higher phage titres, as has been described previously[52]. Our analysis focused on bacteriophages, while we likely also transferred eukaryotic viruses. However, as only 0.044 ± 0.3% (median: 0%) of reads mapped to such viruses, we could not ascertain whether these had an effect. In addition, our sequencing method focused on dsDNA phages, thus missing ssDNA phages, which can be as abundant as dsDNA phages in the human gut[7,8]. Therefore, we suggest future studies include ssDNA viruses, as well as dsRNA and ssRNA viruses for a more comprehensive analysis of the gut phage community. The small sample size and large heterogeneity did not allow for any post hoc sex-based analyses. Finally, since we only included Dutch European subjects, the generalisability of our results to other populations is limited.

Besides above-mentioned suggestions for future FFT studies, future research should focus on targeting specific bacteria with phages to get a better mechanistic understanding of how bacterial communities are changed upon phage predation and how these changes could affect disease phenotypes. One example of specific phages targeting pathogenic bacteria is the phage cocktail developed to treat recurrent *Clostridioides difficile* infections[53]. Another interesting target are the *Lactobacillaceae* that are thought to produce ethanol and thereby contribute to non-alcoholic fatty liver disease (NAFLD)[54]. It should be noted that such precision therapy might be very efficient at clearing a specific pathogen, but will unlikely restore any underlying microbial dysbiosis. Therefore, a combination of endogenous phages to modulate a complete microbiome should be further studied, e.g., by matching donors and recipients based on their phageome and bacteriome composition, respectively.

In conclusion, we performed a double-blind, randomised, placebo-controlled trial in which we safely administered a faecal filtrate to human individuals with MetSyn. We provide evidence that gut phages from a healthy donor can transiently alter the gut microbiota of recipients. This study provides a critical basis for follow-up studies, which should better match donors and recipients based on their bacteriome and phageome composition.

## Methods
### Study design
We set up a prospective, double-blinded, randomised, placebo-controlled intervention study that was performed in our academic hospital, the Amsterdam University Medical Centres location AMC in the Netherlands. After passing screening, 24 subjects with MetSyn were randomised to receive a sterile FFT from a lean healthy donor or a placebo transplant. Prior to the intervention and after 28 days at follow-up, subjects underwent an OGTT to assess their glucose metabolism. In addition, a week prior to one week after intervention, subjects monitored their blood glucose using a flash glucose monitoring device (Freestyle Libre). Faecal samples were collected at multiple timepoints between baseline and follow-up to study dynamic changes in the

microbiome. Finally, during every study visit a medical exam was conducted in addition to blood plasma collection to assess the safety of the intervention. Figure 1a provides a schematic overview of the study.

## Study subjects

Study participants were all European Dutch, overweight (body mass index (BMI) ≥ 25 kg/m$^2$) subjects between 18 and 65 years of age and had to meet the National Cholesterol Education Program (NCEP) criteria for the metabolic syndrome[31]. Both male and female participants were included in the study and sex was self-reported. Main exclusion criteria were the use of any medication, illicit drug use, smoking, or alcohol abuse in the past 3 months, as well as a history of cardiovascular, gastrointestinal, or immunological disease. Supplementary Table 3 summarises all in- and exclusion criteria and Supplementary Fig. 1 provides and overview of the recruitment of study participants.

## Donor screening

Faeces donors were lean healthy European Dutch subjects who were thoroughly screened according to the guidelines of the European FMT Working Group[55]. Screening of potential donors was performed in a stepwise manner as previously published[32]. Briefly, potential donors first completed an extensive screening questionnaire. If they passed this stage, their faeces were screened for pathogenic parasites. When negative, several faecal samples were screened for presence of pathogenic bacteria, viruses and multidrug resistant organisms (MDROs), as well as the level of calprotectin. Donors screened after May 2020 were additionally screened for severe acute respiratory syndrome coronavirus 2 (SARS-CoV-2)[56]. In addition, blood was collected for serological testing and to screen for an abnormal liver or renal function or an impaired immunity. When donors passed this screening, they were allowed to donate faeces for a period of 6 months. Supplementary Table 4 lists the specific in- and exclusion criteria for faeces donors. Every two months, active donors underwent a short rescreening, which included, among others, screening for MDROs and SARS-CoV-2. In addition, before every donation, donors had to complete a shortened questionnaire to confirm their eligibility. We matched donors and recipients based on their sex and whether they have had a prior infection with cytomegalovirus or Epstein–Barr virus.

## Sterile faecal filtrate production and administration

Production of the sterile faecal filtrate started the day before administration to the MetSyn subjects. First, 50 g of stool was collected from a screened donor, which was homogenised with 500 ml sterile saline. Large particles were filtered from the faecal suspension using double sterile gauzes. Most of the bacteria were removed in two subsequent centrifugation steps, in which the suspension was spun for 1 h at 10.000×$g$. Finally, the supernatant was filtered through a sterile 0.2 μm membrane using a tangential flow filtration device (Vivaflow 50). Production of the filtrate from donor stool was performed within 6 h and took, on average, 334 min (SD = 27). The filtrate was stored overnight in a fridge until administration. The production is depicted in Supplementary Fig. 3A.

The sterile faecal filtrate was administered to the patient via a nasoduodenal tube. The day prior to the administration, subjects were asked to clean their bowel using a laxative (Klean-Prep®, Norgine B.V.), which is a standard pre-treatment for FMT procedures in our hospital. Nasoduodenal tubes were placed with the help of a Cortrak®2 enteral access system (Avanos Medical Inc.), making sure the nasoduodenal tube was correctly positioned. The faecal filtrate was slowly infused with a 60 ml syringe, on average 300 ml during a 15-20 min period. Supplementary Fig. 3B provides a schematic overview of the FFT procedure.

During the optimisation of the tangential flow filtration, we quantified the VLP numbers of the faecal filtrates from four different donors, as previously described[57,58]. Briefly, faecal filtrates were concentrated, from which VLPs were isolated with caesium chloride density gradient centrifugation, stained with SYBR Gold and counted by epifluorescence microscopy. Faecal filtrates contained on average 1.25E + 08 VLPs/ml (SD 0.45E + 08), which is in line with previous publications[59,60]. We confirmed the absence of bacteria from the faecal filtrate with a qPCR for the bacterial 16 S rRNA gene as previously described[61], showing a 10$^5$-fold decrease in bacterial DNA (Supplementary Fig. 3C). We further confirmed this by culturing of the faecal filtrate using Biosart® 100 monitors (Sartorius). 100 ml of faecal filtrate was filtered and the cellulose nitrate membranes were incubated on petri dishes with Columbia agar + 5% sheep blood (bioMérieux) for two days at 37 °C under both aerobic and anaerobic conditions. We did not observe any colony-forming units in 100 ml of faecal filtrate.

## Outcomes

The primary outcome was change in glucose metabolism, as determined by the AUC for glucose excursion during the OGTT. Secondary outcomes related to glucose metabolism were changes in fasting glucose, insulin, HOMA-IR and HbA1c between baseline and follow-up after 28 days, as well as changes in glucose variability measured by CGM a week before and after intervention. Other secondary outcomes were the dynamic changes in gut bacteriome and virome populations following FFT or placebo intervention and the comparison of phage composition between lean donors and subjects with MetSyn. Finally, we assessed the safety of the FFT as determined by the occurrence of (serious) adverse events, physical exam and several blood parameters for renal and liver function and inflammation.

## Sample size calculation

Based on previous data from our group in which individuals with MetSyn received an FMT[28,29], and the hypothesis that a faecal phage transplant can be equally effective as a traditional FMT[25–27,30], we assumed a 15% improvement in glucose tolerance upon FFT. With a two-sided 5% significance level and a power of 80%, a sample size of 12 patients per group was necessary, given an anticipated dropout rate of 10%. To recruit 24 individuals with MetSyn, we anticipated a 12-month inclusion period.

## Randomisation

Data were captured with electronic case report forms build in Castor EDC (v2019.2.0-2020.2.25)[62]. In CASTOR, subjects were randomly assigned to an intervention by block randomisation with stratification for age and sex, and block sizes of 4, 6 and 8. The day prior to the intervention, both the faecal filtrate and placebo (sterile saline with brown colour) were prepared and stored overnight. Both the faecal filtrate and placebo looked identical. A randomisation assistant unblinded for the treatment allocation prepared the correct solution for administration and destroyed the other. The investigator administered the allocated treatment in blinded syringes and through an opaque nasoduodenal tube, making sure both participants and the investigator were blinded for the intervention throughout the study.

## Oral glucose tolerance test and biochemical measurements

For the OGTT, overnight fasted subjects ingested a standardised glucose solution (75 g). Blood was drawn from an intravenous catheter at baseline and 15, 30, 45, 60, 90 and 120 min after ingestion. Both blood serum and plasma were aliquoted and stored at −80 °C. From these aliquots we measured glucose and C-peptide, which was performed by the Endocrinology department of the Amsterdam UMC. In addition, blood samples collected at baseline and follow-up were used to measure fasted glucose, insulin, HbA1c and the clinical safety parameters for renal/liver function and inflammation, all measured by the Central Diagnostics Laboratory of the Amsterdam UMC. Results were reported

in the electronic medical record software from EPIC (versions August 2019-November 2020).

## Continuous glucose monitoring

To reduce the study burden and prevent daily finger pricks, we used a continuous glucose monitoring device (Freestyle Libre 1) to monitor blood glucose, which allowed subjects to perform all normal activities while wearing the sensor. Subjects were taught to subcutaneously implant the CGM sensor and were instructed to extract the data from the sensor at least every 8 h. One week prior to the intervention subjects started to monitor their glucose until one week after the intervention. Compliance among participants was good, with a median 100% (range 76–100%) of data correctly collected, during a median period of 14 (range 11–27) days with a median 1350 (range 1043–2617) sensor readings. During that same period, participants were asked to record their diet using an online food diary (Eetmeter v2019-2020) from the Voedingscentrum[63]. At the follow-up visit, data from the CGM scanner were exported and analysed with the previously published CGDA package v0.8.2 for CGM data analysis[64].

## Faeces collection

The day before the intervention and 2, 4, 7, 14 and 28 days thereafter, subjects were asked to collect several faecal samples. Faeces were collected by participant in stool collection tubes, which were directly stored in a freezer at home inside a safety bag. In addition, participants registered the time, date and consistency of the collected faeces according to the Bristol Stool Chart. At the baseline and follow-up visits, these faecal samples were transported to the hospital frozen, where they were directly stored at −80 °C until the end of the study.

## Bacteriome and virome sequencing

To study the bacteriome and virome, we performed whole genome shotgun (WGS) sequencing. From the stored frozen faeces samples, total genomic DNA was extracted using a repeated bead beating method as described previously[33]. Briefly, 250 mg of faeces were weighed in bead-beat tubes, 700 μl of S.T.A.R. buffer (Roche Diagnostics Cat# 03335208001) was added, and samples were homogenised three times using a bead-beater (FastPrep-24™, MP Biomedicals™) set to 5.5 ms for 1 min. Lysates were incubated at 95 °C for 15 min centrifuged for 5 min at full speed (14,000×$g$) at 4 °C, and supernatant was transferred to nuclease-free tubes. The above was repeated once with 300 μl of S.T.A.R. buffer to extract any remaining DNA. DNA was further cleaned from the lysates using the Maxwell® RSC Blood DNA Kit (Promega Cat#ASB1520) according to manufacturer's instructions. Extracted DNA was stored at −20 °C and shipped on dry ice to Novogene (Cambridge, United Kingdom). Libraries for shotgun metagenomic sequencing were prepared using the NEBNext Ultra II Library prep kit (New England Biolabs Cat#E7645L) and sequenced on an Illumina HiSeq instrument with 150 bp paired-end reads and 6 Gb data/sample. Supplementary Figure 3D summarises the sequencing and bioinformatics pipeline used. For both the WGS and VLP sequencing (see below) negative controls were included to check for contamination during DNA extraction and library prep. These negative controls did not yield any measurable DNA after library prep and were therefore not sequenced. No mock communities were included as positive controls in the current sequencing pipeline.

## VLP sequencing

To study phage virions, we isolated the faecal VLP fraction and sequenced dsDNA phages as previously described[19]. Briefly, the VLPs were extracted from 500 mg of faeces using high-speed centrifugation followed by filtration through a 0.45 μm membrane. Any free-DNA debris was digested prior to lysing the VLPs, whereafter the DNA was purified using a two-step phenol/chloroform extraction protocol. Finally, the DNA was purified using the DNeasy Blood&Tissue kit

(Qiagen Cat#69506) according to the manufacturer's protocol. Library preparation was done with the NEBNext Ultra II FS DNA library prep kit (New England Biolabs Cat#E7805L) and the NEBNext Multiplex Oligos for Illumina dual indexes (New England Biolabs Cat#E7600S) according to manufacturer's instructions. Quality and concentration of the VLP libraries were assessed with the Qubit dsDNA HS kit (Thermo-Fisher Cat#Q32854) and with the Agilent High Sensitivity D5000 ScreenTape system (Agilent Technologies). Libraries were sequenced using 2×150 bp paired-end chemistry on an Illumina NovaSeq 6000 platform with the S4 Reagent Kit v1.5, 300 cycles (Illumina Cat#20028312) at the Core Facility Genomics of the Amsterdam UMC.

## Sequence assembly

Sequencing resulted in an average of 21.7 ± 3.5 M reads per WGS sample (median: 22.4 M reads), and 23.6 ± 18.3 M per VLP sample (median: 18.1 M reads). Before assembly, reads belonging to the same participant were concatenated. Adapter sequence removal and read trimming were performed with fastp v0.23.2 (option –detect_adapter_for_pe)[65]. As previously recommended[66], reads were then error corrected with tadpole (options mode=correct, ecc=t, prefilter=2), and deduplicated with clumpify (options dedupe=t, optical=t, dupedist=12000), both from bbmap v38.90 [https://jgi.doe.gov/data-and-tools/bbtools]. High-quality reads from WGS samples were then cross-assembled per participant using metaSPAdes v3.15.5[67] (option–only-assembler). Due to their great complexity, we were unable to assemble some of the VLP samples. We thus assembled these with MEGAHIT v1.2.9[68], which we did for all VLP samples to keep methodological consistency.

## Viral sequence recognition and clustering

To identify viral sequences among the WGS and VLP assemblies, contigs longer than 5000 bp were analysed with virsorter v2.2.3[69] (option –exclude-lt2gene) and checkv v1.0.1[70]. Contigs were taken to be of viral origin if at least one of the following criteria was true: checkv identified at least one viral gene, VirSorter2 gave a score of at least 0.95, VirSorter2 identified at least 2 viral hallmark genes, checkv identified no viral or bacterial genes. In total, we selected 53,204 contigs with at least 1 viral gene, 782 with a virsorter2 score of > 0.95, and 1 with at least 2 viral hallmark genes. The resulting viral sequences were then deduplicated at 100% with bbdupe from bbmap v38.90 (option minidentity=100). This resulted in a non-redundant database of 50,724 viral contigs, which were subsequently clustered at 90% average nucleotide identity (ANI) into viral populations (VPs) using blastn all-vs-all searches with BLAST v2.12.0+[71]. The longest contigs in each VP were further clustered into viral clusters (VCs) by vContact2 v0.11.3[72]. Since the conclusions of the analyses were identical regardless of whether they were performed with VPs or VCs, only VP-level analyses were reported.

## Viral read depth determination

Viral relative abundance was determined by mapping high-quality reads from each sample (i.e., one mapping per participant and time-point) against non-redundant viral sequences with bowtie2 v2.4.2[73]. Following earlier recommendations[74], contigs were considered to be present if at least 75% of their bases were covered by at least 1 read mapped with over 90% ANI. To determine this, reads mapping with less than 90% ANI were removed from alignments with coverm filter v0.6.1 (option –min-read-percent-identity 90 [https://github.com/wwood/CoverM]), and coverage was determined with bedtools genomecov v2.27.1[75] (option -max 1). Read counts per contigs were then determined with samtools idxstats v1.15.1[76], and those with a horizontal coverage of <75% were set to zero. Read counts and contig lengths were summed per VP, and reads per kilobase per million mapped reads (RPKM) values were calculated to take differential contig lengths.

## Bacterial community profiling and binning

Bacterial population compositions of WGS samples were profiled per participant and time point with mOTUs v3.0.3[77]. Binning contigs into metagenome assembled genomes (MAGs) was done per participant. First, high quality reads from each time-point were mapped to cross-assembled contigs of at least 2500 bp with bowtie2 v2.4.2. Read depth tables were then constructed with jgi_summarize_bam_contig_depths v2.15, and contigs were binned with metabat2 v2:2.15[78]. Completion and contamination of putative MAGs were then determined using checkm v1.2.1[79] and, like was previously done[80], MAGs were considered for further analysis if completeness - (5 x contamination) was at least 50. Taxonomy of such MAGs was determined with GTDB-Tk v2.1.1[81] using the R207-v2 database package. This resulted in a database of 3011 MAGs with an assigned taxonomy.

## Determining phage-host links

Viral sequences were linked to bacterial MAGs in two ways. Firstly, if a viral contig was contained within a MAG, it was considered to be a prophage. Secondly, viral contigs were linked to MAGs using CRISPR spacer hits. For this, CRISPR spacer arrays were identified among MAGs using CRISPCasFinder v4.2.20[82]. CRISPR spacers between 20 and 30 bp in length were then matched to viral contigs through a blastn search with BLAST v2.12.0+ (options -task blastn-short). Spacer hits were finally filtered for those with 2 or fewer mismatches, minimising the risk of spurious hits.

## Statistical analyses

All statistical analyses were performed in R v4.2.1. Richness, α-diversities, principal component analysis (PCA), and principal response curves (PRC) were all calculated with the vegan R package v2.6-4[83]. For richness and α-diversity RPKM values were used, while PCAs and PRCs used centred log ratio (clr)-transformed data so as to account for the compositionality of the data[84]. Before clr-transformation, VPs of low abundance and prevalence were removed by removing those with total RPKM of <100 over all samples, as well as those with RPKM values of >20 in less than 10% of samples. Significance levels of PCAs were calculated with a permutational analysis of variance (PERMANOVA) test, as implemented in the vegan R package v2.6-4 and were controlled for age and sex. For the PRC-analysis, the permutest function was used to calculate significance. Both PERMANOVA and permutest used 1000 permutations. p-values were adjusted for multiple testing using the Benjamini–Hochberg approach where necessary. General linear models were constructed with the glmmPQL function from the MASS R package v7.3-58.1 with the age, sex, day, group and day:group as fixed effects and participants as random effect.

## Differential abundance

Differential abundance of VPs among VLP samples on day 2 was determined with ANCOM-BC v1.2.2[85]. Input of ANCOM-BC consisted of the raw read counts summed per VP in each sample, because this method has its own internal data normalisations to account estimated sample fractions. ANCOM-BC was run on VPs with at least 20 reads reported in at least 10% of samples. To account for the relatively small sample sizes, structural zero discovery was turned on but the usage of the asymptotic lower bound turned off[85]. Differential abundance was corrected for the effects of age and sex. ANCOM-BC used multiple-testing correction according to the Benjamini-Hochberg method, with a significance cutoff of 0.05. The number of differentially abundant (DA) VPs was then determined per host species. Enrichment of host species among DA VPs was calculated using a hypergeometric test as implemented in the phyper R function, with the number of DA VPs infecting a given species-1 as q, the total number of VPs in the dataset infecting the same species as m, the total number of VPs with host-m as n, the total number of DA VPs as k, and lower.tail set to FALSE.

## Phage-host interactions

To determine the dynamics of phage-bacterium interaction across the entire population, the change in relative abundance between days 0 and 2, 2 and 28 and 0 and 28 were determined for all VPs with a host and all MAGs with a known phage. The resulting values were then averaged for both VPs and MAGs at the species level, after which Spearman correlation coefficients were calculated.

## Patient and public involvement statement

Patients were involved in the assessment of the grant proposals for this study by the Dutch Diabetes Research Foundation (Diabetes II Breakthrough grant (459001008) and Senior Fellowship (2019.82.004)). Moreover, the patient panel advised on the patient burden of the clinical study. In addition, patients were involved in the ethical approval of this study (as part of the ethics committee). Once the trial results became available, participants were informed of the results with a letter suitable for a non-specialist audience.

## Ethics approval and informed consent statement

This study involves human participants and was approved by the Medical Research Ethics Committee Academic Medical Center Amsterdam (METC 2018_231). Both participants and faeces donors gave informed consent to participate in the study before taking part. The study was registered at the Dutch National Trial Register (NTR) under NL8289 on the 15th of January 2020, while the first patient was included in October 2019. The delay in registration was due to a miscommunication between investigators. When this mistake came to light during the first monitor visit after the first three patients had been included, the study was directly registered at the NTR. This registry does not exist anymore and all data has been added unaltered to the Dutch Trial Register (LTR) under https://clinicaltrialregister.nl/en/trial/26916. While these data are automatically included in the International Clinical Trial Registry Platform (ICTRP), thereby fulfilling the requirement of prospective registration as required by the International Committee of Medical Journal Editors (ICMJE), it was unfortunately no longer possible to adjust the data.

## Reporting summary

Further information on research design is available in the Nature Portfolio Reporting Summary linked to this article.

## Data availability

The sequencing data generated in this study have been deposited in the European Nucleotide Archive database under accession code: PRJEB60691. The data are freely available without restriction. Source data of figures and tables are provided in the Source Data file. The R207-v2 database package of GTDB-Tk is available from: https://data.gtdb.ecogenomic.org/releases/release207/207.0/auxillary_files/gtdbtk_r207_v2_data.tar.gz. Individual participant data that underlie the results reported in this article, after deidentification, are freely available in the Source Data file. The study protocol is included with the publication. All of the individual participant data collected during the trial, after deidentification, are available upon request. Methodological sound proposals with clear aims should be directed to h.j.herrema@amsterdamumc.nl. To gain access, data requestors will need to sign a data transfer agreement. Data will be available up to five years following article publication. Source data are provided with this paper.

## Code availability

The following programmes were used for data collection: Castor EDC (v2019.2.0-2020.2.25), EPIC (versions August 2019-November 2020), Voedingscentrum Eetmeter (v2019-2020), Freestyle Libre 1 software. The following software and codes were used for data analysis: R v4.2.1, CGDA v0.8.2, fastp v0.23.2, bbmap v38.90, metaSPAdes v3.15.5, MEGAHIT v1.2.9, virsorter v2.2.3, checkv v1.0.1, BLAST v2.12.0+,

vContact2 v0.11.3, bowtie2 v2.4.2, coverm filter v0.6.1, bedtools v2.27.1, samtools v1.15.1, mOTUs v3.0.3, jgi_summarize_bam_contig_depths v2.15, metabat2 v2:2.15, checkm v1.2.1, GTDB-Tk v2.1.1, CRISPCasFinder v4.2.20, vegan R package v2.6-4, MASS R package v7.3-58.1 and ANCOM-BC v1.2.2. In the "Methods" section any specific settings are described. Codes and scripts are publicly available online and from published literature. No custom software was developed for this project.

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

## Acknowledgements

We want to thank Aline Fenneman, Klaartje de Bruin and Melanie Benard for their help with stool donor recruitment and screening. In addition, we

thank Veera Houttu, Ulrika Boulund, Kim Dzobo and Torsten Scheithauer for performing the randomisation to guarantee blinding of the investigator and volunteer. We would also like to thank Ana Gerós, Yannick van Schajik and Stephanie Handana for their help in setting up the faecal filtrate production and optimisation. We are grateful for the help of the gastroenterology department of the Amsterdam UMC, location AMC, for their support with the Cortrak®2 enteral access system. We also like to acknowledge the Microbiota Centre Amsterdam for their help with the DNA isolation of the faecal samples and the Genomics core of the Amsterdam UMC, location AMC, for their advice on the whole genome shotgun sequencing. Finally, we are most grateful for the volunteers who participated in this clinical pilot study. Researchers were supported by various grants. K.W. was supported by a Novo Nordisk Foundation CAMIT grant 2018 (28232) to M.N. and a Diabetes II Breakthrough grant (459001008) to H.H.; P.A.dJ. and T.P.M.S. were supported by DDRF Senior fellowship (2019.82.004) to H.H.; I.A. was supported through a Le Ducq consortium grant (17CVD01) to M.N.; M.N. was supported by a personal ZONMW-VICI grant 2020 (09150182010020) and a Le Ducq consortium grant (17CVD01). H.H. was supported by a Senior Fellowship of the Dutch Diabetes Research Foundation (2019.82.004). The funders had no role in the study design, the collection, analysis and interpretation of data, the writing of the report and the decision to submit the article for publication.

## Author contributions

H.H. and M.N. conceived the research idea and designed the study; K.W. performed the clinical study; K.W. and T.P.M.S. processed the samples in the laboratory; P.A.dJ., K.W. and I.A. performed data analysis; K.W. and P.A.dJ. wrote the first draft of the manuscript; All authors contributed to manuscript revision, read and approved the submitted version.

## Competing interests

M.N. is founder and scientific advisor of Caelus Health, however none of this bears any relevance to the content of the current paper. K.W., P.A.dJ., T.P.M.S., I.A., E.M.K. and H.H. report no conflict of interest.
