## [Peer Review File · Nature Communications]

REVIEWER COMMENTS

Reviewer #1 --Fecal virome transplantation / Fecal filtrate transplantation / Microbiome / Metabolic disease-- (Remarks to the Author):

Overall the manuscript by Wortelboer et al is interesting and well-written. In all fairness this reviewer also have admit having read a previously submitted version of the manuscript and I find the manuscript much improved. In the following a number of concerns, questions and comments that could/should be addressed are lined out.

Line 87-88: The transplant also contain eukaryotic viruses, so the safety does not only relate to the transfer of phages, but the entire virome. Consider to rephrase.

Somewhere around line 93: What was the primary endpoint for the RCT?

Line 95-96: Was the 24 subjects based on a power calculation? (I later found out that yes, a power calculation was carried out - this should be mentioned around here in the manuscript)

Line 120-122: Could this be due to transfer of eukaryotic/human viruses? This deserve to be discussed somewhere in the manuscript.

Line 276-278: The argument is a bit difficult to follow. Please rephrase.

Line 291-294: Phage genomes are rather small (1-2% of a bacterial genome), so unless the WGS is sequenced a lot deeper than the VLP fraction it is expected that a lot of low abundant phages will not be "caught" when doing WGS. Please add a comment on this.

Line 300-301: Or perhaps there is no difference? Consider rephrasing.

Line 407-408: I believe the primary outcome is so important, that it deserves to be mentioned in the main part of the paper

Line 418-424: In the discussion it is suggested to carry out RCTs with more participants than the 2*12 used here. Does this mean that the assumptions of the power calculation were wrong? If yes, which numbers/assumptions should future power calculations be based on? Please expand the discussion of this.

Line 465: That only dsDNA phages are sequenced deserve to be mentioned in the Discussion (when discussing limitations of the study)

Line 488-489: Line 163: Virome assembly using metaSpades etc. would probably be better than MEGAHIT - any particular reason why MEGAHIT was chosen beyond "complexity"?

Line 531-537: CRISPR-matching is definitely a good idea. WiSH host-prediction could also aid in this.

Overall, sequencing: Was any mock communities used as positive control for phages and bacteria? Same question for negative controls.

I miss an estimated VLP count of the FFTs – a low titer may be the cause of weak effect. As stated, it is a numbers game.

Reviewer #2 -- Biostatistics / informatics / genomics / clinical trials-- (Remarks to the Author):

I found the manuscript to be interesting and clearly written. The topic is important and potentially exciting. I have several queries and concerns about the results reported and their significance as detailed below:

1. For adverse events, shouldn't the test be comparing 8 out of 12 versus 2 out of 12? With Fisher's exact test I believe you would get a significant p-value ($p=0.03$). Even without this, I would see this as a significant proportion in the treatment group and potentially of concern? Is it fair to say then that the treatment is well tolerated?

2. I am not sure I understand why the “time between 3.9-10 mmol/L glucose” difference is interesting and worth highlighting in the abstract. Firstly, the legend of the figure correctly notes that this is not statistically significant after accounting for the multiple tests done here. Secondly, this difference seems to be within the FFT group (day 0 vs day 28) rather than between the control and FFT groups (day 28) as one would want to see.

3. Line 173: Is the reduction statistically significant?

4. Line 149, 178 etc - please specify the test that was used.

5. Line 180-193: None of the results here are statistically significant, as a minimum bar to conclude that any of these observations are interesting to note.

6. What is the coefficient shown in figure 4A? Was this p-value corrected for multiple testing?

7. For figure 4D are the p-values corrected? What is the legend for the *'s?

8. In figure 5A, the correlation coefficients are quite small and visually they do not appear to be meaningful. I wouldn't call these effects pronounced. There could be other technical explanations for this slight shift. For e.g. could the difference not be explained by the fact that the treatment group has VLPs introduced into their gut microbiome, while in both groups bacteria are depleted due to laxative treatment? Given current analysis and observations, I think the evidence is not strong enough to make the claim that FFTs induce an antagonistic phage-microbe interaction.

9. I appreciated reading the discussion section which was well written and captured many of the points that I believe are important.

Reviewer #3 --Virome / Phageome-- (Remarks to the Author):

The manuscript by Wortelboer and colleagues describes a double-blind, randomised placebo-controlled study on the effect of filtered faecal transplantation (FFT) on metabolic syndrome in a small Dutch cohort. The key finding is that the FFT was safe with no serious adverse effects. A small effect on glucose variability is reported, but no other measures were significant between the placebo and FFT group at the end of the study period. Faecal samples were collected at different time points across the study and investigated for their bacteriome and the dsDNA phageome. These analyses showed differentially abundant viral populations between placebo and FFT, with hypothesized transient antagonistic phage-microbe interactions after FFT.

As far as I'm able to assess, the study set up and data processing are of high quality. The small cohort size means that only large effects can be observed (power of 80% at 5% significance).

My main concern regarding the data for this manuscript is the absence of raw data reporting. The data availability statement "Data are available upon reasonable request" is not acceptable for an open access publication. The raw sequence reads of the metagenomes and viromes, and the assemblies or viral populations should be made publicly available, as well as the raw data for aggregate tables reported in the paper.

I have two major concerns regarding the evaluation of the data and the conclusions drawn.

Firstly, the authors acknowledge that there is an effect of the laxative that was given before both FFT and placebo. They specifically attribute the drop in insulin levels (lines 138-140) to this intervention. However, there are other observations where the effect of the laxative was not acknowledged. For example, would it be possible that the observed difference in glucose variability (Figure 1F) was due to the laxative and not the FFT?

Secondly, the authors only very briefly acknowledge in the discussion that they cannot rule out an effect of other compounds that are present in the filtrate, such as debris, metabolites and peptides. In my opinion, this deserves much more attention. It is highly likely that the effect of metabolites on the microbiome will be higher than the effect of phages that were not matched with their hosts. When reading the manuscript, the mention of the other compounds come as an afterthought, whereas it is a crucial component of the FFT, and should be investigated further and addressed throughout the manuscript.

In addition to the major comments above, I have a number of more specific comments and corrections that should be addressed.

Figure 3, panels A and C have different day values of the X-axes. Please correct.

Lines 226-228: The authors write that the FFT group was dominated by virulent phage-bacterium interactions, versus temperate or chronic in the placebo group. I disagree with this hypothesis. There is no evidence that the interactions in the FFT group were caused by virulent phages. It is equally likely that these interactions were the result of lytic events caused by temperate phages. The authors show no virus classification or lifestyle analysis that supports their hypothesis.

Can the authors also comment on the effect of the laxative again here?

Lines 480 and on: For the sequence data analysis, I see no mention of human read removal. Was this performed? For patient privacy, this is strongly recommended before storing sequence data.

Line 496: The authors have included contigs as viral if they were predicted, but had no viral or bacterial genes identified by CheckV. Did they check these contigs against other databases, such as human or fungi? Would it be possible that these are not viral contigs but belong to a different kingdom?

Lines 536-537: Allowing for up to 5 mismatches in a spacer match is too much, in my opinion, given the short lengths of these spacers. This will lead to many spurious matches.

RESPONSE TO REVIEWERS

We are thankful to the reviewers for their time spent reading and carefully evaluating our work. (Special thanks to reviewer 1 for accepting to review our work a second time!) We were excited to read the reviewers' overall positive feedback and fair assessment of the challenges and limitations of the study. In the point-by-point rebuttal below, we addressed the comments and helpful suggestions. We also processed these in the revised manuscript, which we feel has been much strengthened by the adjustments made.

Reviewer #1 --Fecal virome transplantation / Fecal filtrate transplantation / Microbiome / Metabolic disease-- (Remarks to the Author):

Overall the manuscript by Wortelboer et al is interesting and well-written. In all fairness this reviewer also have admit having read a previously submitted version of the manuscript and I find the manuscript much improved. In the following a number of concerns, questions and comments that could/should be addressed are lined out.

Line 87-88: The transplant also contain eukaryotic viruses, so the safety does not only relate to the transfer of phages, but the entire virome. Consider to rephrase.

We rephrased to the following (lines 96-98): *"In this double-blind, randomised, placebo-controlled pilot study, we provide proof of concept that a faecal filtrate from lean healthy donors containing gut virions can be safely administered to MetSyn recipients."*

Somewhere around line 93: What was the primary endpoint for the RCT?

The primary endpoint for the study was glucose metabolism, as determined by the total area under the curve (AUC) for glucose excursion during the oral glucose tolerance test (OGTT). We have included one line on the primary outcome in a first explanatory section within the results (lines 105-109): *"To study whether an FFT could induce a similar effect on glucose metabolism as an FMT, we set up a prospective double-blind, randomised, placebo-controlled pilot study. Changes in glucose metabolism were determined by the total area under the curve (AUC) for glucose excursion during an oral glucose tolerance test (OGTT), the primary outcome."* Since outcomes are described in the methods section as well, we kept this rather short.

Line 95-96: Was the 24 subjects based on a power calculation? (I later found out that yes, a power calculation was carried out - this should be mentioned around here in the manuscript)

We have included a line on the sample size in the first explanatory section within the results, but kept this rather short since this is already described in the methods section. (lines 109-111) *"Based on previous data from our group and the hypothesis that a faecal phage transplant can be equally effective as an FMT, a sample size of 12 patients per group was deemed necessary."*

Line 120-122: Could this be due to transfer of eukaryotic/human viruses? This deserve to be discussed somewhere in the manuscript.

We agree with the reviewer that eukaryotic viruses could theoretically induce mild gastrointestinal complaints as seen within our study. However, since the donors were screened for the presence of human viruses, we do not think these had an effect on the observed side effects. In any case, we mapped all reads to the eukaryote viral sequences found in RefSeq and using the same mapping procedures and coverage cut-offs, we found that an average of 0.044% of reads from VLP samples mapped to eukaryotic viruses. Without the 75% horizontal coverage cut-off advised previously (10.7717/peerj.3817), 0.24% of reads mapped to eukaryotic viruses. We therefore conclude that we cannot say anything meaningful about this.

We have added the following to the results section (lines 140-145): *“Besides the transferred faecal phages, these adverse events could theoretically be induced through the transfer of eukaryotic or human viruses. However, as only $0.044 \pm 0.3\%$ (median: 0%) of reads mapped to such viruses, we could not ascertain whether these had an effect. To minimize negative effects from eukaryotic viruses, healthy stool donors were thoroughly screened for presence of known pathogenic viruses prior to donation”*.

Line 276-278: The argument is a bit difficult to follow. Please rephrase.

We rephrased the sentence to make the argument more clear (lines 306-309): *“These virulent interactions in the FFT group could be the result of donor phages infecting and lysing the bacteria from the recipient. On the other hand, the introduction of novel donor phages could have induced the replication of existing prophages, thereby leading to more virulent interactions.”*

Line 291-294: Phage genomes are rather small (1-2% of a bacterial genome), so unless the WGS is sequenced a lot deeper than the VLP fraction it is expected that a lot of low abundant phages will not be “caught” when doing WGS. Please add a comment on this.

While WGS samples do indeed have a lower “resolution” for low-abundance phages, we also believe that this may be a reflection of the completely different populations represented in the VLP vs WGS sequencing data (see Supplementary Figure 2C). We added the following to the text (lines 324-326): *“This difference may have been caused by either the absence of low-abundance VPs in the WGS sequencing data due to their relatively smaller sizes, or a difference in community dynamics between VLP and WGS phageomes.”*

Line 300-301: Or perhaps there is no difference? Consider rephrasing.

Indeed, this could be a potential explanation why we do not find an effect. However, it is impossible to say whether the absence of a significant effect is because of no effect or because of the small sample size and large heterogeneity. Therefore we have rephrased the first sentence to the following (lines 333-336): *“Although we did not find an overall effect on glucose metabolism, it is not possible to assess whether the FFT intervention was insufficient to alter the glucose metabolism or whether the effect is obfuscated by the small sample size and large heterogeneity within the MetSyn study population.”*

Line 407-408: I believe the primary outcome is so important, that it deserves to be mentioned in the main part of the paper

We have added a paragraph to the first section of the results shortly explaining the study design and primary outcome. (lines 107-109) *“Changes in glucose metabolism were determined by the total area under the curve (AUC) for glucose excursion during an oral glucose tolerance test (OGTT), the primary outcome.”*

Moreover, the primary outcome is described in the results as the first finding under “FFT improved glucose variability”. To make it more clear, we have added a phrase that this was the primary outcome. (lines 154-155) *“Prior to the intervention and after 28 days at follow-up, subjects underwent an OGTT to assess their glucose metabolism (figure 1A), which was the primary outcome of the study.”*

Line 418-424: In the discussion it is suggested to carry out RCTs with more participants than the 2*12 used here. Does this mean that the assumptions of the power calculation were wrong? If yes, which numbers/assumptions should future power calculations be based on? Please expand the discussion of this.

The sample-size for the current study was based on the assumption that the FFT would be as effective as an FMT in improving glucose metabolism. This was not the case. However, to repeat the power calculation for a future study, there should be an improvement in the primary outcome, i.e. glucose AUC during OGTT. Unfortunately, we found a slight, non-significant increase in AUC upon intervention in both groups. This could be seen as a natural progression of MetSyn, but we speculate that this was caused by the laxative pre-treatment, which also reduced the fasting insulin levels and associated HOMA-IR values at baseline (as compared to day of screening, see reviewer 3). We have added this to the discussion under limitations (lines 336-342)

Line 465: That only dsDNA phages are sequenced deserve to be mentioned in the Discussion (when discussing limitations of the study)

We have added a comment on this to the limitations section (lines 359-361): *“In addition, we focussed on dsDNA phages. Although these phages form the majority of gut phages, for future studies it would also be interesting to include the ssDNA, dsRNA and ssRNA viruses.”*

Line 488-489: Line 163: Virome assembly using metaSpades etc. would probably be better than MEGAHIT - any particular reason why MEGAHIT was chosen beyond "complexity"?

SPAdes is indeed often reported to result in longer contigs, however, several of our VLP samples simply could not be assembled using SPAdes with the computational resources available to us. Because of this, we were forced to use MEGAHIT instead. For the sake of methodological consistency, and to preclude the possibility that we introduced any methodological bias, we chose to assemble all VLP samples with the same sequencer (MEGAHIT).

We edited the methods to reflect this (lines 555-557): *“Due to their great complexity, we were unable to assemble some of the VLP samples. We thus assembled these with MEGAHIT v1.2.9 [61], which we did for all VLP samples to keep methodological consistency.”*

Line 531-537: CRISPR-matching is definitely a good idea. WiSH host-prediction could also aid in this.

WiSH has a reported accuracy of only 63%. Furthermore, we previously found that WiSH predicted a bacteroidetes host for phage contigs that share high homology to Gammaproteobacteria across over 50% of the genome (doi: 10.3390/v11121085). We thus doubt the value of using WiSH for host prediction over the much more reliable CRISPR-spacer and prophage-to-MAG matching.

Overall, sequencing: Was any mock communities used as positive control for phages and bacteria? Same question for negative controls.

We have added the following statement to the methods section (lines 524-528): *“For both the WGS and VLP sequencing (see below) negative controls were included to check for contamination during DNA extraction and library prep. These negative controls did not yield any measurable DNA after library prep and were therefore not sequenced. No mock communities were included as positive controls in the current sequencing pipeline.”*

I miss an estimated VLP count of the FFTs – a low titer may be the cause of weak effect. As stated, it is a numbers game.

We agree with the reviewer that numbers are of relevance in FFT or FVT studies. We did, however, not count VLPs in the transplants. This was due to the logistics of the study and IRB/pharmacy requirements of the FFT. We were not allowed to concentrate the FFTs, which made counting very difficult. In addition, we could store the FFT's no longer than overnight at four degrees. FFT was then administered *as is* the next morning. Since participants did not come in all at once (two per month on average), we could not standardize the number of VLPs administered per participant.

Nevertheless, we have a good idea on VLP numbers in FFTs that were prepared in the described preparation pipeline. Table 1 below shows numbers and images of VLPs in four CsCl-concentrated FFTs prepared according to our tangential flow filtration pipeline on four different days. We show that the average number of VLPs per ml filtrate is 1,25E+08 and rather consistent.

Table 1. VLP count in FFT. VLPs (from FFT prepared as described in our manuscript) were concentrated using CsCl and visualized using SYBR Gold (according to PMID: 19300441 and PMID: 17406585, resp.).

Donor #	VLPs/ml
H11	1,23E+08
H12	8,06E+07
P12	1,86E+08
P13	1,07E+08
Average	1,25E+08

We have added the following to the methods section (lines 442-447): *During the optimisation of the tangential flow filtration, we quantified the VLP numbers of the faecal filtrates from four different donors, as previously described [PMID: 19300441 and PMID: 17406585]. Briefly, faecal filtrates were concentrated, from which VLPs were isolated with caesium chloride density gradient centrifugation, stained with SYBR Gold and counted by epifluorescence microscopy. Faecal filtrates contained on average 1.25E+08 VLPs/ml (SD 0.45E+08), which is in line with previous publications."*

Reviewer #2 -- Biostatistics / informatics / genomics / clinical trials-- (Remarks to the Author):

I found the manuscript to be interesting and clearly written. The topic is important and potentially exciting. I have several queries and concerns about the results reported and their significance as detailed below:

1. For adverse events, shouldn't the test be comparing 8 out of 12 versus 2 out of 12? With Fisher's exact test I believe you would get a significant p-value ($p=0.03$). Even without this, I would see this as a significant proportion in the treatment group and potentially of concern? Is it fair to say then that the treatment is well tolerated?

For adverse events, we compared the number of participants who reported adverse events that could have been related to the intervention. Therefore the comparison was 6/12 vs 2/12. Two participants in the FFT group reported two adverse events that occurred at the same time (e.g. nausea and bloating). In addition to the number of participants, we also compared the total number of adverse events, taking the relatedness into account. The differences between both groups were not statistically significant. We do acknowledge that a larger portion of individuals experienced side effects in the FFT group. However, these side effects were all mild gastrointestinal complaints (such as nausea, bloating, watery stools), were short lived and resolved within a couple of days after the administration. In addition, similar side effects are known for the traditional FMT, which is also well tolerated by patients. Therefore, in our opinion, the treatment was well tolerated.

2. I am not sure I understand why the “time between 3.9-10 mmol/L glucose” difference is interesting and worth highlighting in the abstract. Firstly, the legend of the figure correctly notes that this is not statistically significant after accounting for the multiple tests done here. Secondly, this difference seems to be within the FFT group (day 0 vs day 28) rather than between the control and FFT groups (day 28) as one would want to see.

We have removed the glucose variability from the abstract and included a line on the overall effects on glucose metabolism, since this was the primary outcome of the study. (lines 37-38) *“The FFT was well-tolerated and safe, while the overall changes in glucose metabolism were similar in both groups.”*

3. Line 173: Is the reduction statistically significant?

This was not significant. We added these p-values to the text (lines 197-204): *“Interestingly, in both groups the bacterial richness and α -diversity reduced slightly the first days after the intervention, which was resolved by day 14 to 28, though these decreases were non-significant ($p > 0.05$, Wilcoxon signed rank test). A similar non-significant trend was observed for the richness and diversity of the WGS phageome, which consists mainly of prophages that could have been depleted with their bacterial hosts. In contrast, the richness of the VLP phageome increased slightly by day 2 in both groups, while the α -diversity decreased only in the placebo group, albeit non-significant ($p > 0.05$, Wilcoxon signed rank test).”*

		Day 0 v Day 2		Day 0 v Day 4	
		richness	diversity	richness	diversity
Bacteria	FFT	0.235	0.566	0.273	0.319
	Placebo	0.175	0.198	0.435	0.198
WGS	FFT	0.487	0.695	0.266	0.443
	Placebo	0.219	0.178	0.178	0.101
VLP	FFT	0.551	0.977		
	Placebo	0.908	0.142		

4. Line 149, 178 etc - please specify the test that was used.

We have specified the statistical tests that were used in the text.

5. Line 180-193: None of the results here are statistically significant, as a minimum bar to conclude that any of these observations are interesting to note.

We anticipated effects of the intervention on the viral and bacterial community. Hence we feel the analyses we show would be the first choice of analyses in any study that performs the type of intervention we did (e.g., FMT, FVT, FFT). Not reporting the results because the data are not statistically significant would be reporting bias. We added a small writing to the results section. (lines 207-208) *“Since we expected transfer of donor phages to the recipients, we looked at the abundance of phages shared between donor and recipient before and after the FFT.”*

6. What is the coefficient shown in figure 4A? Was this p-value corrected for multiple testing?

The coefficient is the canonical coefficient of treatment reported by the PRC analysis. These p-values were adjusted for multiple testing by the Benjamini-Hochberg method. We added a mention to this to the figure caption. (lines 1004-1006) *“A) Principal response curve showing how the FFT group differs from the placebo (set to zero) in the bacteriome, bulk-derived phageome (WGS), and phage virion (VLP) composition. The coefficient is the canonical coefficient of treatment and significance in dispersion over time and at each separate time-point was calculated with permutation tests, corrected for multiple testing by the Benjamini-Hochberg method.”*

7. For figure 4D are the p-values corrected? What is the legend for the *'s?

These p-values were adjusted for multiple testing by the Benjamini-Hochberg method. We added a mention to this to the figure caption. (lines 1010-1014) *“D) Bacterial host species of which the phages are enriched among differentially abundant VPs. The first column shows the number of differentially abundant VPs, the second the total number of VPs linked to a given host in the dataset, and stars show the level of significance after testing for enrichment with a hypergeometric test, adjusted for multiple testing by the Benjamini-Hochberg method.”*

8. In figure 5A, the correlation coefficients are quite small and visually they do not appear to be meaningful. I wouldn't call these effects pronounced. There could be other technical explanations for this slight shift. For e.g. could the difference not be explained by the fact that the treatment group has VLPs introduced into their gut microbiome, while in both groups bacteria are depleted due to laxative treatment? Given current analysis and observations, I think the evidence is not strong enough to make the claim that FFTs induce an antagonistic phage-microbe interaction.

While the coefficients are small, they are significant, and thus noteworthy to us. Because this analysis specifically looks at phage-host pairs and their joint increase/decrease in abundance, we think that these results do reflect the interactions between bacteria and phages. However, we agree that our assertions are hypothetical and have weakened them in the text: (lines 254-256) *“These results could indicate a difference in the ecological dynamics between the two sample groups, where the FFT group was dominated by lytic phage-bacterium interactions, while they were more likely to be lysogenic or chronic in the placebo group.”*

9. I appreciated reading the discussion section which was well written and captured many of the points that I believe are important.

We thank the reviewer for this compliment and hope to have further improved the discussion after processing reviewers comments and suggestions.

Reviewer #3 --Virome / Phageome-- (Remarks to the Author):

The manuscript by Wortelboer and colleagues describes a double-blind, randomised placebo-controlled study on the effect of filtered faecal transplantation (FFT) on metabolic syndrome in a small Dutch cohort. The key finding is that the FFT was safe with no serious adverse effects. A small effect on glucose variability is reported, but no other measures were significant between the placebo and FFT group at the end of the study period. Faecal samples were collected at different time points across the study and investigated for their bacteriome and the dsDNA phageome. These analyses showed

differentially abundant viral populations between placebo and FFT, with hypothesized transient antagonistic phage-microbe interactions after FFT.

As far as I'm able to assess, the study set up and data processing are of high quality. The small cohort size means that only large effects can be observed (power of 80% at 5% significance).

My main concern regarding the data for this manuscript is the absence of raw data reporting. The data availability statement "Data are available upon reasonable request" is not acceptable for an open access publication. The raw sequence reads of the metagenomes and viromes, and the assemblies or viral populations should be made publicly available, as well as the raw data for aggregate tables reported in the paper.

We believe the editor has reached out to this reviewer to state that the data are accessible. We agree with the reviewer that the raw data should be publicly available, especially for an open access publication. At the time of submission, we uploaded the raw sequencing data to the European Nucleotide Archive database under accession code PRJEB60691. Unfortunately, the processing took a little bit longer and thus we provided the editor with a SURF Drive link to the raw data that could be shared with reviewers. At the time of acceptance, the data should be accessible at the ENA.

I have two major concerns regarding the evaluation of the data and the conclusions drawn. Firstly, the authors acknowledge that there is an effect of the laxative that was given before both FFT and placebo. They specifically attribute the drop in insulin levels (lines 138-140) to this intervention. However, there are other observations where the effect of the laxative was not acknowledged. For example, would it be possible that the observed difference in glucose variability (Figure 1F) was due to the laxative and not the FFT?

We do indeed believe the drop in insulin levels are related to the laxative used in the study, since this effect is present in both groups, and the lower levels are only apparent at the baseline visit (after the laxative and prior to FFT) and not at the screening and follow-up visit (see figures below). The laxative effect is probably short-lived, as long as subjects remain fasted. Like the lower insulin values, we also see a trend towards a lower glucose AUC at baseline compared to day 28 in both groups. Unfortunately, we did not perform an OGTT during the screening and thus it is impossible to say if the lower glucose AUC at baseline is a laxative effect, or whether the increase in glucose AUC by day 28 is a reflection of the natural MetSyn disease progression. We have added a line on this to the limitations section in the discussion (lines 340-342): *"The increased glucose AUC could be seen as natural progression of MetSyn, but we speculate that this was caused by the laxative pre-treatment, which also reduced the fasting insulin levels and associated HOMA-IR values at baseline."*

For the glucose variability, which was measured with a continuous glucose monitor we deem it unlikely that the observed difference within the FFT group was caused by the laxative. Firstly, the CGM measurements were collected one week before and one week after the intervention and the data from the day before and day of the intervention were not included in the data analysis (because of the fasting, OGTT and FFT intervention). Therefore the CGM measurements are less influenced by short term influences such as the laxative. Secondly, the improvement in the FFT group is only visible within the FFT group, not within the placebo group.

Secondly, the authors only very briefly acknowledge in the discussion that they cannot rule out an effect of other compounds that are present in the filtrate, such as debris, metabolites and peptides. In my opinion, this deserves much more attention. It is highly likely that the effect of metabolites on the microbiome will be higher than the effect of phages that were not matched with their hosts. When reading the manuscript, the mention of the other compounds come as an afterthought, whereas it is a crucial component of the FFT, and should be investigated further and addressed throughout the manuscript.

Metabolites, peptides and other compounds could have a potential effect on the microbiome besides the faecal phages. Theoretically, a propagating and evolving entity can have a lasting effect on the microbiome, while this is less likely for specific metabolites or compounds. This is why we mainly focussed on the viral compartment. Moreover, since it was not within the scope of our current study, we did not have the resources for metabolomics or proteomics and thus did not investigate this further.

Ideally, we would further clean the faecal filtrate and concentrate just the phage fraction to use as intervention in this study. Also, a more interesting and relevant control than sterile saline as placebo would be the use of a heat-inactivated filtrate that still contains the metabolites and other compounds without any viable phages. However, we were only allowed to minimally process the faecal suspensions by the IRB to limit the risks for the participants and thus we chose for an FFT and compared it with sterile saline as placebo.

We have added the following to the introduction (lines 87-94): *“To study the effect of faecal phages on glucose metabolism, comparing a clean and concentrated faecal virome transplant with a phage-inactivated transplant would be most desirable. Unfortunately, the IRB only allowed us to minimally process the faecal suspension that is usually used for FMT, so we chose an FFT approach. We were*

hence not able to remove components other than bacteria from the filtrate. However, since phages are self-propagating entities with presumed longer effects on the microbial ecosystem than a single administration of metabolites, peptides or debris, we considered it justified to use the FFT to study phage-bacteria interactions and subsequent effects on glucose metabolism.”

In addition to the major comments above, I have a number of more specific comments and corrections that should be addressed.

Figure 3, panels A and C have different day values of the X-axes. Please correct.

The X-axis of panel A and C should be from day 0 to 28. The figures have been adjusted accordingly.

Lines 226-228: The authors write that the FFT group was dominated by virulent phage-bacterium interactions, versus temperate or chronic in the placebo group. I disagree with this hypothesis. There is no evidence that the interactions in the FFT group were caused by virulent phages. It is equally likely that these interactions were the result of lytic events caused by temperate phages. The authors show no virus classification or lifestyle analysis that supports their hypothesis.

Can the authors also comment on the effect of the laxative again here?

With this statement, we meant to say that the FFT group seems to have more lytic events (whether due to virulent or temperate phages) than the placebo. As such we accidentally mislabelled lytic as virulent and lysogenic as temperate. We thus changed the text to (lines 254-256): *“These results could indicate a difference in the ecological dynamics between the two sample groups, where the FFT group was dominated by lytic phage-bacterium interactions, while they were more likely to be lysogenic or chronic in the placebo group.”*

Both groups received the laxative. It is unlikely that these differences between the groups are induced by the laxative.

Lines 480 and on: For the sequence data analysis, I see no mention of human read removal. Was this performed? For patient privacy, this is strongly recommended before storing sequence data.

The data set has been uploaded to the ENA before the removal of human reads, which is not conflicting with the informed consent form signed by the participants.

Line 496: The authors have included contigs as viral if they were predicted, but had no viral or bacterial genes identified by CheckV. Did they check these contigs against other databases, such as human or fungi? Would it be possible that these are not viral contigs but belong to a different kingdom?

While we did use this as a rule, no viral contigs were actually selected because of this reason. In total, we selected 53,204 contigs with at least 1 viral gene, 782 with a *virsorter2* score of > 0.95, and 1 with at least 2 viral hallmark genes. For clarity we added this to the text in the methods: (lines 563-565) *“In total, we selected 53,204 contigs with at least 1 viral gene, 782 with a virsorter2 score of > 0.95, and 1 with at least 2 viral hallmark genes.”*

Lines 536-537: Allowing for up to 5 mismatches in a spacer match is too much, in my opinion, given the short lengths of these spacers. This will lead to many spurious matches.

We fully agree with the reviewer, and this was due to a typo. We actually used a cut-off of 2 or less, which we think minimizes the risk of spurious hits. Increasing the number of allowable mismatches would actually diminish the effect in Figure 5A: with five mismatches the statistics are FFT R2 -0.1 p 0.00024, placebo R2 0.13 p 0.00077. We amended this in the methods (line 606).

We notably did spot an error in our code relating to Figure 4D/E, where we inadvertently did use 5 mismatches. Using 2 mismatches, we see slightly different results (specifically, *Eisenbergiella*, *Bacteroides fragilis*, and *Bacteroides cellulosyliticus* are not significantly enriched). As the *Bacteroides* are enriched at the genus level, we added genus-level analysis to this figure.

** See Nature Portfolio's author and referees' website at www.nature.com/authors for information about policies, services and author benefits.

This email has been sent through the Springer Nature Tracking System NY-610A-NPG&MTS

Confidentiality Statement:

This e-mail is confidential and subject to copyright. Any unauthorised use or disclosure of its contents is prohibited. If you have received this email in error please notify our Manuscript Tracking System Helpdesk team at <http://platformsupport.nature.com> .

Details of the confidentiality and pre-publicity policy may be found here <http://www.nature.com/authors/policies/confidentiality.html>

Privacy Policy | Update Profile

REVIEWER COMMENTS

Reviewer #1 (Remarks to the Author):

I thank the authors for all the updates to the manuscript, which have improved it a lot. I am also satisfied with their responses to my concerns. There is only one remark, where I am not quite happy with the response:

My previous comment: Line 465: That only dsDNA phages are sequenced deserve to be mentioned in the Discussion (when

discussing limitations of the study)

Author response: We have added a comment on this to the limitations section (lines 359-361): "In addition, we focussed

on dsDNA phages. Although these phages form the majority of gut phages, for future studies it would

also be interesting to include the ssDNA, dsRNA and ssRNA viruses."

My new comment/remark. For many individuals ssDNA phages are as abundant as dsDNA phages in the gut (though they are missed by the Nextera library prep that many uses, so some papers underestimate them). Please update text to reflect that ssDNA and dsDNA together form the vast majority of gut phages

Reviewer #2 (Remarks to the Author):

1. Similar to the reporting of changes in other sections, even when they are not statistically significant, it seems appropriate that the section on safety note an increase in the number of adverse events in the FFT group (statistically significant, $p=0.03$), and an increase in the number of subjects with adverse events in the FFT group (marginally significant, $p=0.08$). The argument that FFT is safe and well-tolerated seems to rest on the observation that no serious adverse events were seen. However, given a cohort size of 12 subjects, this seems to be an underpowered claim that is not well-established by this study.

2. Given the nature of statistical testing, it is more appropriate to say "... while the overall changes in glucose metabolism were not significantly different in both groups".

3. As noted in my previous review, no result in the section "Increase in new phages independent of the intervention" seems to have passed statistical significance. While I appreciate the need to avoid reporting bias", the claim in the title and also in the sentence "These results seem to indicate that the phageomes were perturbed in both the placebo and FFT groups." therefore do not follow from the statistical analysis that has been shown.

4. Was multiple testing correction done with the ANCOM-BC analysis? It is still not clear what p-value corresponds to a single star vs two stars in Figure 4D-E. How are the results in Figure 4 consistent with the results in Figure 3 where donor phage abundances don't change?

5. For the phage-host correlation analysis, was the compositionality effect accounted for? It is well known that it is easy to obtain "significant correlations" when the number of datapoints being considered is large. This is why it is common to also look at the effect size i.e. the correlation coefficient, where correlation coefficients < 0.3 are typically not considered interesting, correlation coefficients > 0.3 but < 0.7 indicate weak correlation, while coefficients > 0.7 are considered strong correlations. Thus the claim that "overall effect of the FTT on phage-host interactions seemed pronounced" is not supported by the results present in this work. Overall, I do not see where the following claim in the abstract is established: "...which coincided with more virulent phage-microbe interactions". Other potential explanations for the very weak correlations observed such as the fact that the treatment group has VLPs introduced into their gut microbiome, while in both groups bacteria are depleted due to laxative treatment, have not been accounted for.

Reviewer #3 (Remarks to the Author):

All my concerns have been addressed. This is an exciting study that will be of interest to all in the field. Congratulations to the authors!

RESPONSE TO REVIEWER COMMENTS

We are thankful to the reviewers for their time spent reading and carefully evaluating our revised work. We were excited to read the reviewers' overall positive feedback. In the point-by-point rebuttal below, we addressed the remaining comments and helpful suggestions, which we processed in the revised manuscript.

Reviewer #1 (Remarks to the Author):

I thank the authors for all the updates to the manuscript, which have improved it a lot. I am also satisfied with their responses to my concerns.

We thank the reviewer for their compliments.

There is only one remark, where I am not quite happy with the response:

My previous comment: Line 465: That only dsDNA phages are sequenced deserve to be mentioned in the Discussion (when discussing limitations of the study)

Author response: We have added a comment on this to the limitations section (lines 359-361): "In addition, we focussed on dsDNA phages. Although these phages form the majority of gut phages, for future studies it would also be interesting to include the ssDNA, dsRNA and ssRNA viruses."

My new comment/remark. For many individuals ssDNA phages are as abundant as dsDNA phages in the gut (though they are missed by the Nextera library prep that many uses, so some papers underestimate them). Please update text to reflect that ssDNA and dsDNA together form the vast majority of gut phages

We have rewritten the lines in the discussion to reflect the above (lines 360-363):

"In addition, our sequencing method focused on dsDNA phages, thus missing ssDNA phages, which can be as abundant as dsDNA phages in the human gut^{7,8}. Therefore, we suggest future studies include ssDNA viruses, as well as dsRNA and ssRNA viruses for a more comprehensive analysis of the gut phage community."

Reviewer #2 (Remarks to the Author):

We thank the reviewer for their thorough comments and suggestions.

1. Similar to the reporting of changes in other sections, even when they are not statistically significant, it seems appropriate that the section on safety note an increase in the number of adverse events in the FFT group (statistically significant, $p=0.03$), and an increase in the number of subjects with adverse events in the FFT group (marginally significant, $p=0.08$). The argument that FFT is safe and well-tolerated seems to rest on the observation that no serious adverse events were seen. However, given a cohort size of 12 subjects, this seems to be an underpowered claim that is not well-established by this study.

In contrast to what the reviewer points out, the number of adverse events in the FFT group (16) is only slightly larger compared to the placebo group (13). It is unclear to us how the p-value of 0.03 was obtained from only these two numbers. If we take the number of likely and possibly related adverse events vs non-treatment related ones, a Fisher's exact test gives a p of 0.1142 ($p=0.49$ in only likely treatment-related adverse events are taken). Furthermore, the difference in subjects with adverse events is 2 of 12 in the Placebo group vs 6 out of 12 in the FFT group, which gives a p-value of 0.19 (Fisher's exact test). We acknowledge that there were more adverse events that are possibly or likely related to the intervention in the FFT group compared to the placebo group, though this was not significant. This was already stated in the manuscript, but we now highlighted this in the text more clearly (lines 137-142):

"Compared to the placebo group, more subjects in the FFT group reported adverse events (AEs) that were likely or possibly related to the intervention (six vs two subjects, $p = 0.19$, Fisher's exact test), reporting in total more AEs (eight vs two AEs, $p = 0.11$, Fisher's exact test). The adverse events after FFT were in general mild gastrointestinal complaints, such as diarrhoea, constipation, bloating, and nausea, which started in the days after the intervention (median 1 day, range 0 - 36) and disappeared after several days (median 3 days, range 0 - 27)."

The argument that the FFT is safe and well-tolerated is based on the mildness and short duration of the adverse events. These adverse events were expected and are comparable to the traditional FMT, which is an experimental treatment that has been found to be generally safe and well-tolerated. Moreover, as we point out in the discussion, an FFT has less risks compared to the traditional FMT due to a lower risk of transferring unknown pathogenic bacteria.

2. Given the nature of statistical testing, it is more appropriate to say "... while the overall changes in glucose metabolism were not significantly different in both groups".

We were not completely sure to which line the reviewer was referring. We did change a line in the abstract (lines 37-38):

"The FFT was well-tolerated and safe, while the overall changes in glucose metabolism were not significantly different in both groups."

3. As noted in my previous review, no result in the section "Increase in new phages independent of the intervention" seems to have passed statistical significance. While I appreciate the need to avoid reporting bias", the claim in the title and also in the sentence "These results seem to indicate that the phageomes were perturbed in both the placebo and FFT groups." therefore do not follow from the statistical analysis that has been shown.

The statement that the phageomes are perturbed is based on the observed increase in new phages in both groups, which is significant, while the difference between the groups is not significant. Thus, we speculated that this increase in new phages could indicate perturbed phageomes in both groups.

In contrast, the claim in the title about the altered phage-microbe dynamics is not derived from these results, but rather from the part thereafter where we show an altered phage composition after FFT, which coincided with altered phage-microbe interactions. Nevertheless, on advice from the editor we toned down the title of the manuscript to:

"Phage-microbe dynamics after sterile faecal filtrate transplantation in individuals with metabolic syndrome: a double blind, randomised, placebo-controlled clinical trial assessing efficacy and safety".

4. Was multiple testing correction done with the ANCOM-BC analysis? It is still not clear what p-value corresponds to a single star vs two stars in Figure 4D-E. How are the results in Figure 4 consistent with the results in Figure 3 where donor phage abundances don't change?

ANCOM-BC has built-in multiple-testing correction. To make this clearer, we added the following to the methods (lines 634-635):

"ANCOM-BC used multiple-testing correction according to the Benjamini-Hochberg method, with a significance cutoff of 0.05."

The stars in Figure 4 use the standard encoding, which we did not mention by mistake. We apologize for this and added the explanation to the figure legend of Figure 4:

*"significance levels are: * <0.05, ** <0.01, *** <0.001, **** <0.0001."*

Figure 3 shows that donor shared phages overall don't change in abundance, but that does not mean that the treatment/addition of foreign phages did not have indirect effects on the gut virome, such as those depicted in Figure 4, as the gut virome is a very complex system.

5. For the phage-host correlation analysis, was the compositionality effect accounted for? It is well known that it is easy to obtain "significant correlations" when the number of datapoints being considered is large. This is why it is common to also look at the effect size i.e. the correlation coefficient, where correlation coefficients < 0.3 are typically not considered interesting, correlation coefficients >0.3 but <0.7 indicate weak correlation, while coefficients >0.7 are considered strong correlations. Thus the claim that "overall effect of the FFT on phage-host interactions seemed pronounced" is not supported by the results present in this work. Overall, I do not see where the following claim in the abstract is established: "...which coincided with more virulent phage-microbe interactions". Other potential explanations for the very weak correlations observed such as the fact that the treatment group has VLPs introduced into their gut microbiome, while in both groups bacteria are depleted due to laxative treatment, have not been accounted for.

This analysis uses relative abundances because we believed that increases between days 0 and 2 would still be increases when using clr-transformed data. Indeed, when we repeated the analysis with clr-transformed data, the FFT-related correlation became stronger (FFT R=-0.38, p=2.2e-16 and Placebo R=0.15, p = 2.2e-16). Since this evidently did not affect the outcome of the analysis, we chose to maintain the analysis as it is currently in the manuscript. We did, however, add the following to the results (lines 257-258):

"These findings were unaltered when employing clr-transformed data."

As to whether the correlation is interesting: to our knowledge, coefficients <0.3 are not considered uninteresting, but weak, while 0.3-0.7 is considered medium, and >0.7 is considered strong. While our correlation is thus weak, it is definitely significant. Merely because it is weak, one cannot argue that it is uninteresting: weak correlations can certainly point to an important effect, especially in the highly complex gut microbiome. We would like to leave the subjective judgement on whether the correlation is interesting to the reader.

On the third point: we don't understand what the reviewer is implying here. The alternate explanation given by them is that one group received an FFT while the other did not, which is certainly true. But this fact alone cannot explain our observation, as we don't see an increase in all

phages, but more commonly those infecting bacteria that decrease in abundance. After all, our analysis looks at specific phage-host pairs and their change in abundance in the first 2 days. The most reasonable explanation for an increase in the abundance of a phage and a concomitant decrease in abundance of its host over the course of 2 days is that they had an interaction. After all, phages can only increase in abundance by infecting a bacterium. And the type of phage-host interaction that leads to an increase in phage abundance in tandem with a decrease in bacterial abundance is most likely a virulent one. Thus, we would argue that an increase in virulent phage-host interactions during the first 2 days in the FFT group is the easiest explanation of the observed effect. The offered alternate explanation would assume that no interactions took place between the phages and bacteria over the course of 2 days, that certain phages flushed through the system while others did not, and that this happened to coincide with the growth and depletion of their bacterial hosts. This seems to us a highly unlikely explanation.

Reviewer #3 (Remarks to the Author):

All my concerns have been addressed. This is an exciting study that will be of interest to all in the field. Congratulations to the authors!

We thank the reviewer for their compliments.